# Double Horizon Model-Based Policy Optimization

**Akihiro Kubo**[1,2,*]                                          *kubo-a@sys.i.kyoto-u.ac.jp*

**Paavo Parmas**[3]                                          *paavo.parmas@weblab.t.u-tokyo.ac.jp*

**Shin Ishii**[1,2]                                          *ishii@i.kyoto-u.ac.jp*

[1] *Advanced Telecommunications Research Institute*
[2] *Kyoto University*
[3] *The University of Tokyo*
[*] *Corresponding Author*

**Reviewed on OpenReview:** *https://openreview.net/forum?id=HRvHCdO3HM*

## Abstract

Model-based reinforcement learning (MBRL) reduces the cost of real-environment sampling by generating synthetic trajectories (called rollouts) from a learned dynamics model. However, choosing the length of the rollouts poses two dilemmas: (1) Longer rollouts better preserve on-policy training but amplify model bias, indicating the need for an intermediate horizon to mitigate distribution shift (i.e., the gap between on-policy and past off-policy samples). (2) Moreover, a longer model rollout may reduce value estimation bias but raise the variance of policy gradients due to backpropagation through multiple steps, implying another intermediate horizon for stable gradient estimates. However, these two optimal horizons may differ. To resolve this conflict, we propose Double Horizon Model-Based Policy Optimization (DHMBPO), which divides the rollout procedure into a long "distribution rollout" (DR) and a short "training rollout" (TR). The DR generates on-policy state samples for mitigating distribution shift. In contrast, the short TR leverages differentiable transitions to offer accurate value gradient estimation with stable gradient updates, thereby requiring fewer updates and reducing overall runtime. We demonstrate that the double-horizon approach effectively balances distribution shift, model bias, and gradient instability, and surpasses existing MBRL methods on continuous-control benchmarks in terms of both sample efficiency and runtime. Our code is available at https://github.com/4kubo/erl_lib.

## 1 Introduction

Reinforcement learning (RL) is a framework for finding an optimal policy in sequential decision-making applications through trial-and-error interactions with an environment. Model-based RL (MBRL), which learns a dynamics model of an environment in a data-driven manner, leverages the learned model to generate synthetic samples, thereby reducing costly real-environment interactions. However, if the model is imperfect, long synthetic rollouts can accumulate significant errors—known as model bias—especially as the rollout length increases. Additionally, data collected under a poorly performing policy may differ from that encountered by the current policy, resulting in a distribution shift in both policy evaluation (Hallak and Mannor, 2017) and policy improvement (Liu et al., 2019). Several MBRL methods address these issues by tailoring the length and the usage of the model's predictions.

Broadly, these successful approaches can be grouped into two types. The first type employs differentiable rollouts (Deisenroth and Rasmussen, 2011; Amos et al., 2021; Hafner et al., 2019; Hansen et al., 2024) or leverages model derivatives of predictions (Zhang et al., 2023). By backpropagating through a few model-predicted transitions, these methods compute a model-based value expansion (MVE) estimator (Feinberg

et al., 2018), which sums the on-policy rewards for intermediate states with the estimated return from the terminal state (or a critic's prediction). As the MVE estimator refines value estimation, the policy can be improved using more accurate on-policy value estimates even with a lower update-to-data (UTD) ratio—i.e., fewer policy parameter updates per environment step. This lower UTD ratio also helps suppress the increase in computation time (Hiraoka et al., 2022). However, extending differentiable rollouts in conjunction with the reparametrization trick (Kingma and Welling, 2013) can inflate the variance of the policy gradients (Parmas et al., 2018), thereby limiting the effective length of these training rollouts. In this work, we refer to the policy gradient computed via differentiable rollouts as the "value gradient" (Fairbank and Alonso, 2012; Heess et al., 2015; Zhang et al., 2023), distinguishing it from the gradient estimated by a purely model-free algorithm (Sutton et al., 1999; Silver et al., 2014).

Meanwhile, another line of MBRL research—the model-based policy optimization (MBPO) algorithm (Janner et al., 2019)—generates one-step transitions that are treated as if they were obtained from the real environment. Starting from states stored in the replay buffer (which contains states, actions, and rewards), MBPO rolls out a few steps with the learned model to generate synthetic states that more closely match the current policy's state distribution. MBPO has demonstrated excellent empirical performance on the OpenAI Gym benchmark (Brockman et al., 2016). Although the MBPO formalization and theoretical analysis do not necessarily support the benefits of model-generated on-policy data, we interpret its empirical success as evidence that optimizing the policy over an on-policy state distribution can improve learning in practice.

In this work, we propose combining these two ideas: a "distribution rollout" (DR) that approximates the on-policy state distribution (inspired by the model rollouts in implementation of MBPO algorithm), and a "training rollout" (TR) that produces an MVE estimator by exploiting differentiable transitions. However, the optimal horizons for DR and TR may differ. A longer DR helps maintain on-policy samples but also increases model bias, implying an intermediate horizon to minimize distribution shift. In contrast, a longer TR can reduce value estimation bias while raising the variance of the policy gradients, suggesting another intermediate horizon for stable gradient estimates.

Therefore, we propose using different rollout horizons for DR and TR—a long horizon for DR and a short horizon for TR. Since we perform rollouts with different horizon lengths, we refer to this method as the Double Horizon MBPO (DHMBPO) algorithm. In DHMBPO, the TR begins from on-policy states provided by the long DR, enabling us to maintain a short TR horizon while still ensuring accurate value estimation and effective policy improvement. Figure 1 illustrates the concept and key differences between DR and TR.

In experiments on continuous-control benchmarks (Section 4), we demonstrate that DHMBPO not only surpasses existing MBRL methods in sample efficiency but also achieves lower runtime due to a reduced UTD ratio (Hiraoka et al., 2022). Notably, DHMBPO achieved comparable sample efficiency to the state-of-the-art MACURA (Frauenknecht et al., 2024) algorithm on the Gymnasium (Towers et al., 2023) tasks while requiring only one-sixteenth of the runtime on average, all using a shared set of hyperparameters (see A). The remainder of this paper reviews the necessary background (Section 2), details our approach (Section 3), and presents empirical evaluations that highlight the benefits of combining these two complementary rollouts.

## 2 Background

### 2.1 Notation and Problem Setting

In this work, we consider a Markov decision process (MDP) with entropy regularization (Geist et al., 2019). At each environment step $t$, an agent in state $s = s_t$ selects an action $a = a_t$ according to a stochastic policy $\pi_\theta(a \mid s_t)$ parameterized by $\theta$, transitions to the next state $s_{t+1}$ based on the transition probability $p(s \mid s_t, a_t)$, and receives a reward $r_t = r(s_t, a_t)$. Given a discount factor $\gamma \in [0, 1)$ for weighting future rewards, a distribution $\mu(s)$ from which initial states $s_0$ are drawn, and a parameter $\alpha \geq 0$ for the entropy bonus, the objective is to find a policy that maximizes

$$J(\pi_\theta) := \mathbb{E}_{s,a \sim \rho_{\pi_\theta} \pi_\theta} \left[ r(s, a) - \alpha \log \pi_\theta(a|s) \right], \tag{1}$$

where $\rho_{\pi_\theta}$ is the discounted state visitation distribution under $\pi_\theta$, defined as $\rho_{\pi_\theta}(s) := (1 - \gamma) \sum_{t=0}^{\infty} \gamma^t p(s_t = s | \mu, \pi_\theta)$ with $p(s_t = s | \mu, \pi_\theta)$ representing the probability of observing

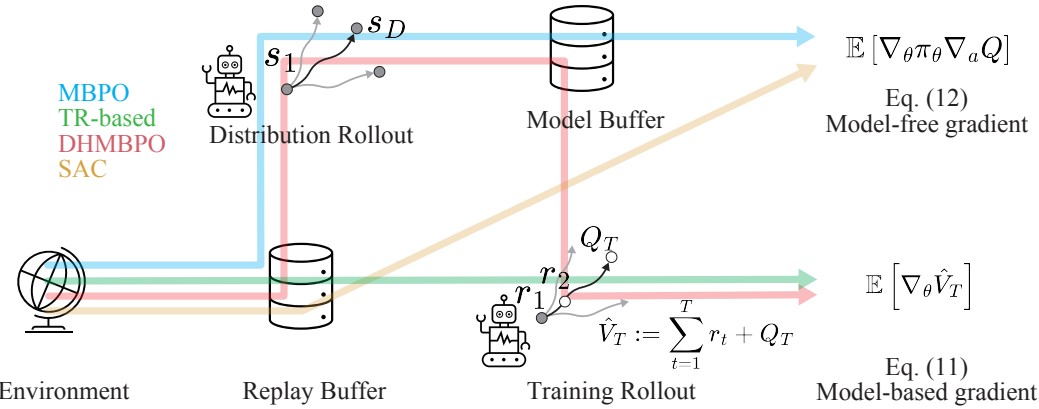

Figure 1: An overview of four approaches to policy gradient estimation: MBPO, TR-based methods, DHMBPO, and an off-policy model-free method. **(1) MBPO (blue lines)**: Starts from states stored in the replay buffer and runs a distribution rollout (DR) with horizon $D$, generating synthetic samples that are then stored in a model buffer. A model-free policy gradient is computed over these samples. **(2) TR-based (green)**: Methods that directly use states from the replay buffer and perform a short-horizon training rollout (TR) with horizon $T$. A model-based gradient estimate—the value gradient—is computed. **(3) DHMBPO (red, proposed)**: Combines both DR and TR. It first applies a DR on samples from the replay buffer to obtain on-policy states, which serve as the initial states for a short-horizon TR. The value gradient is then computed from the TR. **(4) Model-free (yellow)**: Represents a fully model-free algorithm that relies solely on replay-buffer data and computes a model-free gradient estimate.

state $s$ at time $t$ and $p(s_0 = s|\mu, \pi_\theta) := \mu(s)$ . Moreover,

$$V_\alpha^\pi(s) := \mathbb{E}\left[\sum_{t=0}^\infty \gamma^t \left(r(s_t, a_t) - \alpha \log \pi_\theta(a_t|s_t)\right)\Big|s_0 = s\right], \tag{2}$$

is a soft value function, and the objective function (1) is rewritten as $J(\pi) = \mathbb{E}_{s\sim\mu}[V_\alpha^{\pi_\theta}(s)]$. The expectation in eq. (2) is taken over the stochastic trajectory $(s_0, a_0, s_1, \cdots)$ drawn from its distribution $\Pi_{t=0}^\infty \pi(a_t|s_t)p(s_t|\mu, \pi_\theta)$.

We use an associated soft $Q$-function

$$Q_\alpha^\pi(s, a) := r(s, a) + \gamma \mathbb{E}_{s'\sim p(\cdot|s,a)}[V_\alpha^\pi(s')]. \tag{3}$$

We assume that, for all $s \in \mathcal{S}$, $V_\alpha^\pi(s) = \mathbb{E}_{a\sim\pi(\cdot|s)}[Q_\alpha^\pi(s, a) - \alpha \log \pi(a|s)]$.

If $\alpha = 0$, eq. (2) is equivalent to the objective of an unregularized MDP, i.e., $\mathbb{E}[\sum_{t=0}^\infty \gamma^t r(s_t, a_t)]$. Hence, in what follows we do not strictly differentiate between the entropy-regularized MDP and the unregularized MDP; for simplicity, we write $V^\pi(s)$ for the value function and $Q^\pi(s, a)$ for the $Q$-function.

## 2.2 Model-based Actor-Critic Methods

In this study, we consider a deep model-based actor-critic architecture. Specifically, we have neural networks for a policy $\pi_\theta(a|s)$ as an actor, a critic $Q_\phi(s, a)$ that approximates $Q^\pi(s, a)$, and a dynamics model $p_\psi(s'|s, a)$ along with a reward model $r_\psi(s, a)$. The parameters $\theta$, $\phi$, and $\psi$ denote the weights of the respective neural networks.

During critic learning, for an input $(s, a)$, a target signal $\hat{y}$ is computed, and the critic $Q_\phi$ is updated to minimize the squared difference between its output and $\hat{y}$. Meanwhile, the actor $\pi_\theta$ is updated to maximize the sample average (sampled from a given state distribution) of the estimated value $V^{\pi_\theta}(s) \approx \mathbb{E}_{a\sim\pi_\theta(\cdot|s)}[Q_\phi(s, a) - \alpha \log \pi_\theta(a|s)]$. Following existing works (Janner et al., 2019; Hafner et al., 2019; Hansen et al., 2022; 2024), we perform alternating updates of the critic and actor within a policy optimization loop. This alternation is explicitly shown in the for loop (line 8) of Algorithm 1.

## 2.3 Model-Based Policy Optimization (MBPO)

The MBPO algorithm (Janner et al., 2019) applies the policy optimization procedure from SAC (Haarnoja et al., 2018) to fictitious samples generated via model rollouts. Here, we refer to these rollouts as DR. To execute DR, we randomly select initial states from the replay buffer and then simulate $D$ steps using the learned model. The resulting trajectories are stored in a "model buffer", $\mathcal{D}_m^D$, which is separate from the replay buffer.

The critic $Q_\phi$ is updated to minimize:

$$\mathcal{L}_D(\phi) \coloneqq \mathbb{E}_{(s,a,r,s')\sim\mathcal{D}_m^D}\left[\left(Q_\phi(s,a) - \hat{y}\right)^2\right], \tag{4}$$

where the target signal is defined as $\hat{y} \coloneqq r(s,a) + \gamma\,\mathbb{E}_{a'\sim\pi_\theta(\cdot|s')}\left[Q_{\bar{\phi}}(s',a') - \alpha\log\pi_\theta(a'|s')\right]$. Here, $Q_{\bar{\phi}}$ is a target critic with parameters $\bar{\phi}$ computed as an exponentially moving average of $\phi$ (Mnih et al., 2015). The notations $(s,a,r,s') \sim \mathcal{D}_m^D$ and $s \sim \mathcal{D}_m^D$ denote independent sampling operations from the model buffer.

The policy is then updated by stochastic gradient ascent on the surrogate loss[1] given by

$$\mathcal{J}_D(\theta) \coloneqq \mathbb{E}_{s\sim\mathcal{D}_m^D}\left[V_{\phi,\theta}(s)\right], \tag{5}$$

where

$$V_{\phi,\theta}(s) \coloneqq \mathbb{E}_{a\sim\pi_\theta(\cdot|s)}\left[Q_\phi(s,a) - \alpha\log\pi_\theta(a|s)\right]. \tag{6}$$

## 2.4 Training Rollout-based Methods

In this work, we refer to methods that employ the MVE estimator (Feinberg et al., 2018) as *TR-based* with a differentiable rollout (TR). The MVE estimator for a state $s$ uses a $T$-step TR trajectory $\tau_T \coloneqq (s_0 = s, a_0, \hat{s}_1, a_1, \cdots, a_{T-1}, \hat{s}_T)$ generated by the policy $\pi_\theta$ and the model $p_\psi$, to compute:

$$\hat{V}_{\phi,\theta,T}(s) \coloneqq \sum_{t=0}^{T-1} \gamma^t\left(r_\psi(\hat{s}_t, a_t) - \alpha\log\pi_\theta(a_t|\hat{s}_t)\right) + \gamma^T\,\hat{V}_\phi(\hat{s}_T), \tag{7}$$

where $\hat{V}_\phi(\hat{s}_T) \coloneqq Q_\phi(\hat{s}_T, a_T) - \alpha\log\pi_\theta(a_T|\hat{s}_T) \approx \mathbb{E}_{a\sim\pi_\theta(\cdot|\hat{s}_T)}\left[Q_\phi(\hat{s}_T, a) - \alpha\log\pi_\theta(a|\hat{s}_T)\right]$. Even when $\hat{V}_\phi(\hat{s}_T)$ is imperfect—due to the critic $Q_\phi$ struggling to keep pace with a rapidly changing policy—the on-policy reward sequence up to $T-1$ steps improves the overall value estimate at $s$ (Feinberg et al., 2018). Although Feinberg et al. (2018) assumes that the reward function is known, learning a reward model $r_\psi$ typically poses fewer challenges than learning a dynamics model or $Q$-function, so similar benefits are expected.

Following Hafner et al. (2019), we utilize the MVE estimator for learning both of the critic and the policy. Specifically, the critic is trained to approximate the $Q$-function $Q^{\pi_\theta}(s,a)$ by minimizing

$$\mathcal{L}_Q(\phi) \coloneqq \mathbb{E}_{(s,a)\sim\mathcal{D}_e}\left[\left(Q_\phi(s,a) - \hat{Q}_{\bar{\phi},\theta,T}(s,a)\right)^2\right], \tag{8}$$

where

$$\hat{Q}_{\bar{\phi},\theta,T}(s,a) \coloneqq r_\psi(s,a) + \sum_{t=1}^{T-1} \gamma^t\left(r_\psi(\hat{s}_t, a_t) - \alpha\log\pi_\theta(a_t|\hat{s}_t)\right) + \gamma^T\hat{V}_{\bar{\phi}}(\hat{s}_T). \tag{9}$$

For the policy, we perform stochastic gradient ascent on the surrogate loss

$$\mathcal{J}_T(\theta) \coloneqq \mathbb{E}_{s\sim\mathcal{D}_e}\left[\hat{V}_{\phi,\theta,T}(s)\right]. \tag{10}$$

The gradient $\nabla_\theta\hat{V}_{\phi,\theta,T}(s)$ is computed by combining the reparametrization (RP) trick (Kingma and Welling, 2013) with backpropagation through the TR in eq. (7). We refer to the gradient $\nabla_\theta\hat{V}_{\phi,\theta,T}(s)$, as the "value gradient field," which is a function of the state $s$. It has been shown that when $T$ becomes large, this process can be prone to gradient variance explosion (Parmas et al., 2018). Hence, in practice, $T$ is often limited to about 5 steps to maintain stable optimization (Amos et al., 2021).

---

[1]We write "surrogate loss" because the reward in eq. (1) is replaced with the value function, akin to the Off-Policy actor-critic framework of Degris et al. (2012).

## 2.5 Policy-Value Gradient

A key subproblem in policy optimization is to estimate the gradient of the policy parameters, $\nabla_\theta \mathcal{J}(\theta)$, for the objective in (1). We assume the stochastic policy is reparametrized so that for a function $y(a)$: $\mathbb{E}_{a \sim \pi_\theta(a|s)}[y(a)] = \mathbb{E}_{\epsilon \sim Z}[y(\pi_\theta(s, \epsilon))]$ where $Z$ is a simple distribution (e.g, Gaussian), and $\pi_\theta(a|s, \epsilon)$ is a deterministic function mapping $(s, \epsilon)$ to $a$. In this work, we estimate the policy gradient via the RP trick. The policy gradient can be expressed as

$$\nabla_\theta \mathcal{J}(\pi_\theta) = \mathbb{E}_{s \sim \mu}[\nabla_\theta V^{\pi_\theta}(s))] = \mathbb{E}_{s \sim \mu, \varsigma \sim Z}[\nabla_\theta Q^{\pi_\theta}(s, \pi(s, \varsigma))] \tag{11}$$

$$= \mathbb{E}_{s \sim \rho^{\pi_\theta}, \varsigma \sim Z}\left[\nabla_\theta a \cdot \nabla_a Q^{\pi_\theta}(s, a)\big|_{a = \pi_\theta(s, \varsigma)}\right], \tag{12}$$

as derived in (Zhang et al., 2023), with the final expression derived by extending the deterministic policy gradient theorem (Silver et al., 2014). Here, we assume $\alpha = 0$, and the notation "·" denotes an element-wise product summed over the action dimension.

The last term (12) considers only a one-step change in action, which enables us to replace $Q^{\pi_\theta}$ with the critic $Q_\phi$ when approximating the partial derivative $\nabla_a Q^{\pi_\theta}(s, a)\big|_{a = \pi_\theta(s, \varsigma)}$. We refer to the resulting estimate, $\nabla_\theta a \cdot \nabla_a Q_\phi(s, a)\big|_{a = \pi_\theta(s, \varsigma)}$ as "a model-free policy gradient field".

On the other hand, the first term is the (stochastic) value gradient, where $\nabla_\theta V^{\pi_\theta}(s)$ captures how changes in future actions affect the value function. In an RL setting where the true transition probability is unknown, a dynamics model is required to estimate this term. The gradient field $\nabla_\theta \hat{V}_{\phi, \theta, T}(s)$ of the MVE estimator (7) can be used for this purpose. However, it is important to note that the terminal term in this gradient is a model-free policy gradient field $\nabla_\theta a_T \cdot \nabla_a Q_\phi(\hat{s}_T, a_T)\big|_{a_T = \pi_\theta(\hat{s}_T, \varsigma)}$.

Additionally, note that the expectation in equation (12) is taken with respect to the discounted state-visitation distribution, whereas, in the first and second terms of (11), it is taken with respect to the initial state distribution. As implied in Zhang et al. (2023), a mixture of the initial state distribution and the discounted state-visitation distribution may be preferable, the terminal term of the value gradient—approximated by the MVE estimator—is a model-free policy gradient field.

## 2.6 Bootstrap ensemble model and its learning

Following Chua et al. (2018); Janner et al. (2019), we use a bootstrap ensemble of probabilistic neural networks, where the outputs are independent multivariate Gaussian distributions. Individual probabilistic models capture aleatoric noise, while the bootstrapping procedure accounts for epistemic uncertainty. The details of the model, including its architecture, objective function, and early stopping, are described in Appendix A (from Section A.1 to Section A.3).

# 3 Double Horizon Model-Based Policy Optimization

This section introduces the Double Horizon Model-Based Policy Optimization (DHMBPO) algorithm, which integrates DR and TR to approximate both the state distribution and value estimation. Before detailing DHMBPO, we highlight the differences between DR-based methods and TR-based methods.

A common procedure for both DR-based and TR-based methods is to alternate between (i) sampling states and actions from a distribution $d(s, a)$ and (ii) updating policy using samples from the estimated policy gradient field.

MBPO samples states and actions from the model buffer $\mathcal{D}_m^D$, which closely approximates an on-policy distribution, and computes the model-free policy gradient field:

$$d(s, a) = \mathcal{D}_m^D \quad \text{and} \quad \nabla_\theta \pi_\theta(s, \varsigma) \cdot \nabla_a Q_\phi(s, a)\big|_{a = \pi_\theta(s, \varsigma)},$$

where, due to DR, sampling from the model buffer can be interpreted as approximating $\rho^{\pi_\theta}$ in (12).

---

**Algorithm 1** Double Horizon Model-Based Policy Optimization

---

**Require:** Models $p_\psi$ and $r_\psi$, actor network $\pi_\theta$, critic network $Q_\phi$, target critic network $Q_{\bar\phi}$, replay buffer $\mathcal{D}_e$, model buffer $\mathcal{D}_m^D$

1: **for** $e = 0, 1, 2, \cdots$ **do**
2:     Interact with the environment using $\pi_\theta$ and add observed transitions to $\mathcal{D}_e$
3:     Fit the models $p_\psi$ and $r_\psi$ to samples from $\mathcal{D}_e$
4:     Sample states from $\mathcal{D}_e$; then perform $D$-step DR and store the generated transitions in $\mathcal{D}_m^D$
5:     **for** $i = 0$ to $L - 1$ **do**                        ▷ Policy optimization loop
6:         Sample states from $\mathcal{D}_m^D$ and execute $T$-step TR
7:         Compute the MVE estimates (7) and (9)
8:         Update $\phi$ to minimize (13) and then update $\theta$ to maximize (14) via eq. (11)
9:     Clear $\mathcal{D}_m^D$

---

TR-based methods use a replay buffer $\mathcal{D}_e$ in place of $d(s,a)$ and employ a value gradient field via the $T$-step MVE estimator, $\nabla_\theta \hat{V}_{\phi,\theta,T}(s)$:

$$d(s,a) \;=\; \mathcal{D}_e \quad \text{and} \quad \nabla_\theta \hat{V}_{\phi,\theta,T}(s).$$

Note that, as discussed in Section 2.5, sampling from a more on-policy distribution is preferable to sampling from the replay buffer.

In DHMBPO, we propose to replace $\rho_{\pi_\theta}(s)$ with a model buffer $\mathcal{D}_m^D$ generated via DR, and to use a value gradient field of the $T$-step MVE estimator:

$$d(s,a) \;=\; \mathcal{D}_m^D \quad \text{and} \quad \nabla_\theta \hat{V}_{\phi,\theta,T}(s),$$

That is, we randomly select samples from the model buffer, similar to MBPO, and for the actor update, we calculate $\nabla_\theta \hat{V}_{\phi,\theta,T}(s)$ using the same procedure as TR-based methods. This approach reduces distribution shift through DR without requiring on-policy and real-environment interactions while improving value estimation by leveraging the MVE estimator from TR.

Concretely, the critic loss function is defined as

$$\mathcal{L}_{D,T}(\phi) \coloneqq \mathbb{E}_{(s,a)\sim\mathcal{D}_m^D}\Big[\big(Q_\phi(s,a) \;-\; \hat{Q}_{\bar\phi,\theta,T}(s,a)\big)^2\Big], \tag{13}$$

and the actor's objective is given by

$$\mathcal{J}_{D,T}(\theta) \coloneqq \mathbb{E}_{s\sim\mathcal{D}_m^D}\Big[\hat{V}_{\phi,\theta,T}(s)\Big]. \tag{14}$$

Algorithm 1 presents the procedure for the DHMBPO algorithm. Here, $L$ denotes the length of an episode, and repeating the policy optimization $L$ times corresponds to a UTD ratio of 1.

Long DR rollouts help maintain on-policy training but increase model bias, suggesting an intermediate rollout length to minimize distribution shift. In contrast, longer TRs can reduce value estimation error but raise the variance of policy gradients, implying another intermediate horizon for stable gradient estimates. We keep TRs short to ensure stable policy gradient estimation and compensate by using longer DRs, thereby leveraging model predictions to achieve adequate value estimation accuracy while enabling effective policy improvement. In this work, we set the DR horizon to 20 steps and the TR horizon to 5 steps.

## 4 Experiments

We validate our DHMBPO on standard continuous control tasks (from Section 4.1 to Section 4.2). We also present results for sensitivity analyses with respect to rollout lengths, including the special case where either DR or TR is entirely removed (Section 4.3 and 4.4), as well as experiments investigating the impact on efficient critic learning (Section 4.5).

### 4.1 Continuous control benchmark tasks

We evaluate the DHMBPO algorithm on a suite of MuJoCo-based (Emanuel et al., 2012) continuous control tasks from Gymnasium (GYM) (Towers et al., 2023) and DMControl (DMC) (Tunyasuvunakool et al., 2020). These tasks range from basic control problems to high-dimensional robot locomotion, enabling us to assess the general effectiveness of our approach.

These experiments aim to answer three key questions:

- Does the combined use of DR and TR enhance sample efficiency?

- Does it facilitate efficient critic learning?

- How does DHMBPO compare to other state-of-the-art MBRL algorithms?

All DHMBPO runs share a common set of hyperparameters, which, along with other implementation details, are described in Appendix A.

**Evaluation Protocol.** After $x$ environment steps, we measure the algorithm's performance using a *test return*. Specifically, the test return for DHMBPO is computed as the sample mean of the cumulative rewards over 10 episodes, whereas some other methods use fewer episodes (see Appendix B for details).

**Performance Visualization and Statistical Analysis.** We illustrate each method's performance on individual tasks by plotting the mean test return over random seeds (solid line) along with its 95% confidence interval (shaded area). To aggregate and compare results across multiple tasks, we utilize `rliable` (Agarwal et al., 2021), which provides various statistical tools, including sample efficiency curves and aggregation metrics.

However, because existing works typically focus on either GYM or DMC (but not both), their performance ranges are not directly comparable. We thus apply task-specific normalization: (i) the number of environment steps is normalized by the maximum step budget for each task, and (ii) the test return is divided by the final performance of a designated baseline. Note that this additional normalization is our adaptation to unify GYM and DMC results. In our GYM comparisons (Figure 2 in Section 4.2), however, we present raw (unnormalized) returns to align with common practice (Chua et al., 2018; Janner et al., 2019; Amos et al., 2021; Frauenknecht et al., 2024).

In Section 4.3, we present an ablation study that isolates the effect of combining DR and TR, and in Section 4.2, we benchmark DHMBPO against several high-performance MBRL algorithms.

### 4.2 Comparison with State-of-The-Art Algorithms

In this section, we compare the proposed DHMBPO algorithm with several high-performance deep MBRL algorithms whose implementations are publicly available.

For the comparison on the GYM suite, we selected five tasks and compared DHMBPO with MBPO (Janner et al., 2019), SAC-SVG(H) (Amos et al., 2021), RP-PGM (Zhang et al., 2023)] and MACURA (Frauenknecht et al., 2024). In addition to MBPO, SAC-SVG(H) serves as an important baseline algorithm that employs the SVG method—a type of TR-based algorithm. The primary difference between DHMBPO without DR and SAC-SVG(H) is that DHMBPO uses deep ensemble models (Lakshminarayanan et al., 2017), whereas SAC-SVG(H) employs a GRU (Cho, 2014) as its deterministic recurrent neural network. Additionally, SAC-SVG(H) learns the critic using one-step TD targets during the TR. RP-PGM was verified using a modified implementation based on SAC-SVG(H), which was adjusted to mitigate the exploding variance issue in policy gradient estimation. MACURA is based on the MBPO algorithm but adaptively adjusts the length of the DR and is reported to achieve state-of-the-art performance on the GYM suite. Furthermore, as a reference, we included the model-free and off-policy SAC algorithm as a comparison target. Although SAC is model-free, its objective function formulation is highly relevant to those optimized by DHMBPO, MBPO, and SAC-SVG(H) algorithms, particularly in that it includes an entropy regularization term.

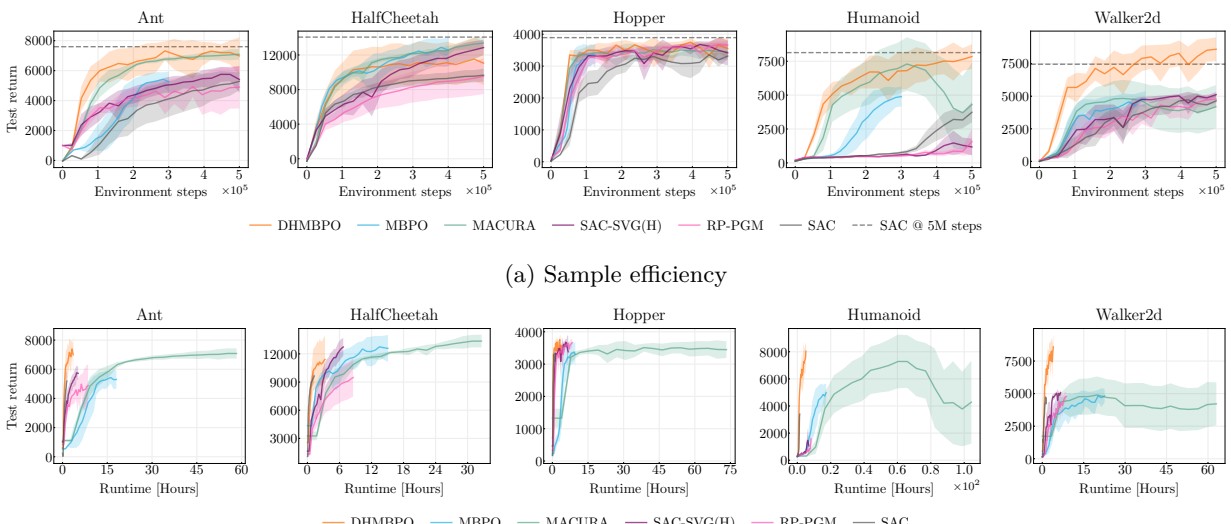

(a) Sample efficiency

(b) Test return curve as a function of runtime at each constant environment step.

Figure 2: Comparison with state-of-the-art methods on GYM tasks in terms of sample efficiency (top) and runtime up to 500K environment steps (bottom). The solid line represents the sample mean over eight random seeds, and the shaded area indicates the 95% confidence intervals. Dashed lines for SAC represent the average test return across 5 random seeds, at ten times the maximum environment step budget used for the model-based methods. The runtime plots show that DHMBPO reaches the highest test returns the fastest, highlighting its efficiency in both sample and runtime cost.

For the comparison on the DMC suite, we selected 18 tasks and compared DHMBPO with TD-MPC2 (Hansen et al., 2024) and Dreamer v3 (Hafner et al., 2023). Both of these are latent model-based actor-critic algorithms that have demonstrated exceptional performance across multiple benchmark suites, including DMC, using common hyperparameters. TD-MPC2 combines representation learning and reward learning, and during behavior execution, it performs model predictive control. Dreamer v3 combines representation learning with the MVE estimator from TR, which is used for training both the critic and the actor.

### 4.2.1   Comparison on GYM Tasks

**Sample efficiency**   The results on GYM tasks are shown in Figure 2a, where the sample mean over eight random seeds is plotted as a solid line and 95% credible intervals are depicted as the shaded area. For MBPO, five random seeds were used. Our experiments confirm that DHMBPO achieved the highest sample efficiency across all tasks even without per-task tuning. This substantial practical performance improvement is arguably the most important contribution of our work. For reference, we include horizontal lines indicating the asymptotic performance of SAC. Specifically, these lines represent the average test return across 5 random seeds, at 5M environment steps, a number of steps equal to ten times the maximum environment step budget used for the model-based methods in each task.

**Runtime Comparison**   For the runtime comparison of DHMBPO, MACURA, and SAC-SVG(H) on GYM tasks, we present the results in Figure 2b and summarize them in Table 1. Each experiment was executed until 500K environment steps on a system configured with 8 NVIDIA RTX A4000 16GB GPUs. Considering the sample efficiency results mentioned earlier, DHMBPO reached 500K steps in less than one-sixteenth of the runtime while achieving the same or higher sample efficiency as MACURA. The primary reason for MACURA's longer runtime is its higher UTD ratio. For example, the UTD ratio for MACURA on `Humanoid` task was 20, while that of DHMBPO (and SAC-SVG(H)) was 1. Although a high UTD ratio can promote higher sample efficiency (Chen et al., 2020; D'Oro et al., 2023), we observed that DHMBPO performed well with a UTD ratio of 1, thereby reducing runtime. In fact, increasing the UTD ratio yielded only

Table 1: Runtimes (in hours), until 500K environment steps, of DHMBPO, MACURA (Frauenknecht et al., 2024) and SAC-SVG(H) (Amos et al., 2021), on GYM tasks and each ratio to DHMBPO's mean runtime.

| Task | DHMBPO | MACURA | SAC-SVG(H) |
|------|--------|--------|------------|
| Ant | 3.6 | 58.3 | 5.2 |
| HalfCheetah | 3.3 | 32.6 | 6.7 |
| Hopper | 3.7 | 73.3 | 6.7 |
| Humanoid | 5.2 | 104.3 | 6.9 |
| Walker2d | 4.0 | 63.0 | 6.6 |
| Ratio | 1.0 | 16.8 | 1.6 |

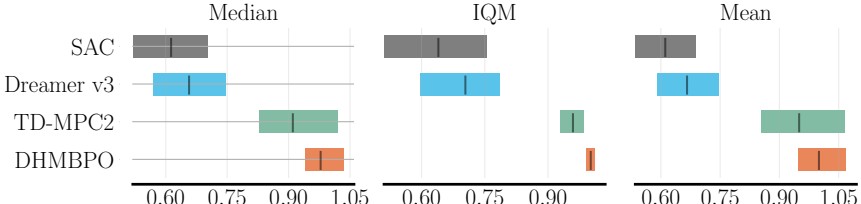

Figure 3: Comparison of aggregation metrics for TD-MPC2 (Hansen et al., 2024), Dreamer v3 (Hafner et al., 2023) and SAC (Haarnoja et al., 2018) on 18 DMC tasks. The metrics are normalized relative DHMBPO's mean scores.

marginal improvements in sample efficiency while significantly increasing execution time (see Figure 20 in Appendix D.5).

### 4.2.2 Comparison on DMC Tasks

Next, in Figure 3, we present the aggregated metrics on DMC tasks. The metrics are about the test return at 50% normalized steps. The three metrics for each method are shown, with the vertical bars representing the metric values and the width of the bars indicating the 95% confidence intervals. The inter quantile mean (IQM) score demonstrates that DHMBPO exhibited significant performance improvements over Dreamer v3 and less variation than TD-MPC2.

Additional task-specific learning curves and runtime results are provided in Section C.

### 4.3 Ablation study

Figure 4 shows the experimental results evaluating the sample efficiency of the proposed DHMBPO algorithm and its variants on a set of 10 continuous control benchmark tasks. We selected five representative tasks from the GYM suite (upper row) and five relatively challenging tasks from the DMC suite (lower row). All compared methods share the same model-learning modules and implementation details, differing only in how DR and TR are combined.

We consider three configurations of DHMBPO corresponding to different (DR horizon, TR horizon) settings: **DHMBPO (20, 5)**, the proposed algorithm, which uses both DR and TR. **DHMBPO without DR (0, 5)**: Initial states are sampled from the replay buffer and used for TR; this setup corresponds to SAC-SVG(H). **DHMBPO without TR (20, 0)**: Initial states are sampled from the model buffer, and the policy optimization procedure follows SAC as described in Section 2.3; this setup corresponds to MBPO.

For each configuration, we ran experiments using 8 different random seeds. The solid lines in Figure 4 represent the mean performance across these seeds, while the shaded areas indicate the 95% confidence intervals.

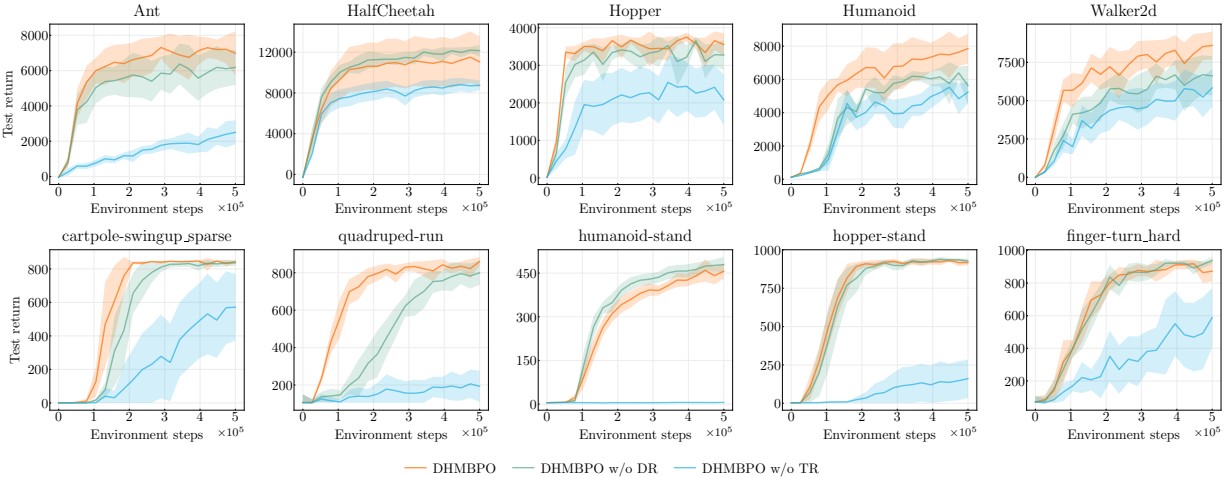

Figure 4: Performance comparison of DHMBPO (orange) against its two ablated variants: DHMBPO without DR (w/o DR, corresponding to SAC-SVG(H), green) and DHMBPO without TR (w/o TR, corresponding to MBPO, blue). The top row shows results on five representative tasks from the GYM suite, and the bottom row shows results on five more challenging tasks from the DMC suite. Each solid line represents the mean test return over 8 random seeds, and the shaded regions denote the 95% confidence intervals. The proposed DHMBPO configuration (DR horizon = 20, TR horizon = 5) consistently outperforms both variants.

The results show that combining DR and TR outperforms using either DR or TR alone. In particular, the DHMBPO configuration (20, 5) achieves higher returns and better sample efficiency than the two reduced variants. For example, the TR-only variant (0, 5), which corresponds to SAC-SVG(H), benefits from improved value estimation; however, its reliance on off-policy samples leads to lower sample efficiency in challenging tasks such as `Humanoid`, `Walker2d`, and `quadruped-run`. Meanwhile, the DR-only variant (20, 0), corresponding to MBPO, can approximate the on-policy state distribution, but the absence of TR's MVE estimation results in ineffective value corrections and notably inferior performance across all tasks. Overall, these results demonstrate that the synergy achieved by combining DR and TR in DHMBPO leads to faster learning than using either technique alone.

We include additional experiments in Appendix D.1 for DHMBPO without TR (corresponding to MBPO) but with different UTD ratios. Increasing the UTD ratio improved sample efficiency (Figure 11) but also resulted in longer execution times (Figure 12). In other words, achieving high sample efficiency without TR requires sacrificing computation cost by raising the UTD ratio. In contrast, DHMBPO achieved high sample efficiency without increasing the UTD ratio (Figure 4), indicating that introducing TR helps reduce runtime cost. Indeed, as shown in Section 4.2, DHMBPO attained sample efficiency comparable to state-of-the-art MBPO-based methods in a shorter execution time.

## 4.4 Hyper-parameter sensitivity

Figure 5 illustrates our sensitivity analysis of the distribution-rollout (DR) and training-rollout (TR) horizon hyperparameters. We evaluated 10 tasks under 8 random seeds, measuring the test return after 500K environment steps. To avoid bias toward any single metric, the figure reports three summary statistics—median, inter-quantile mean (IQM), and mean—along with their 95% confidence intervals.

We chose DR=20 and TR=5 as our baseline. In Figure 5a, we varied only the DR horizon (0, 10, 20, 40), while keeping TR fixed at 5. In Figure 5b, we varied only the TR horizon (1, 3, 5, 7, 9), while keeping DR fixed at 20. The results show that: 1) DR benefits from being relatively long (e.g., 20) compared to short horizons (e.g., 0). 2) TR=5 offers a good balance between sample efficiency and gradient stability. A short TR reduces sample efficiency (similar to "DHMBPO w/o TR" in Figure 4), while excessively long TR can cause gradient norms to explode (see Appendix C.3, Figure 10b), destabilizing learning. Furthermore, in

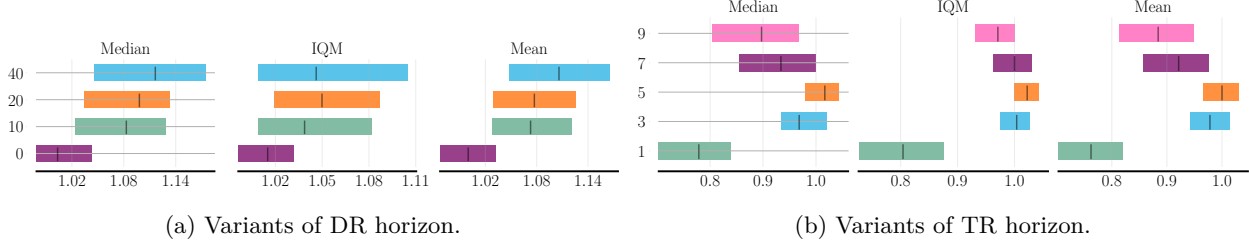

(a) Variants of DR horizon.

(b) Variants of TR horizon.

Figure 5: Aggregation metrics on 10 tasks for (a) variants with different DR horizons and (b) variants with different TR horizons.

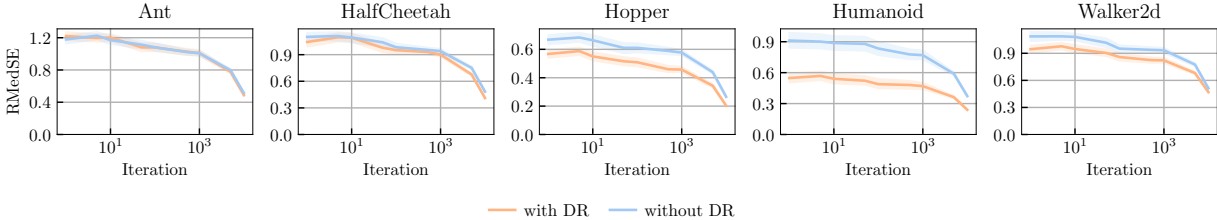

Figure 6: Development of the RMedSE for DHMBPO (orange) and DHMBPO without DR (blue). The $x$-axis represents the number of iterations used to update the critic parameters while all other models remain fixed. The $y$-axis shows the normalized RMedSE with respect to the Monte Carlo return (N=256) from the target environment.

Section D.2, we investigate the trade-off in policy gradient estimation through TR and propose a practical method to determine an optimal TR horizon based on a small number of real samples.

Overall, these findings suggest setting DR to a moderately long horizon and TR to a shorter, carefully chosen horizon to prevent gradient explosion and maintain robust performance.

## 4.5 Efficient Critic Learning

We demonstrate the benefit of DR in critic learning, a more accurate value estimation. We ran DHMBPO for 100K environment steps on five environments from the GYM suite. After training, the learned dynamics model and actor network were fixed, and the critic networks were re-initialized to compare two variants for critic learning: **with DR:** Train the critic using model-generated rollouts stored in a model buffer. **without DR:** Train the critic using only samples from the replay buffer, without additional model-based rollouts.

As ground truth values, we computed a Monte Carlo return $\hat{Q}_g(s,a)$ using 2,048 rollouts in the real environment, for each of 256 randomly sampled state-action pairs $(s,a)$ drawn from the replay buffer. We then computed the MVE estimate $\hat{Q}_{\bar{\phi},\theta,5}(s,a)$ for these 256 pairs, and measured its discrepancy from $\hat{Q}_g(s,a)$ using a normalized root median squared error (RMedSE):

$$E(i) \coloneqq \sqrt{\text{Median}\left[\left(\frac{\hat{Q}_g(s,a) - \hat{Q}_{\bar{\phi},\theta,5}(s,a)}{\hat{Q}_g(s,a)}\right)^2\right]},$$

where $i$ is the critic optimization step. "Median" denotes the median over the selected 256 state-action pairs. For each setup, we repeated the procedure with 32 different random seeds for the critic's initialization and plotted the mean (solid line) and the 95% confidence intervals (shaded area) of $E(i)$ as a function of $i$.

As shown in the last three panels of Figure 6, we observe that the variant *with DR* exhibits a significantly lower estimation error, particularly during the early stages of training. This difference indicates that the MVE estimator more accurately approximates the true value function. Because the policy is continuously updated

throughout training, an early advantage in value estimation can compound and benefit each subsequent optimization step. In the `Humanoid` environment specifically, we believe that this improved value estimation under DR contributes to the notable gains in sample efficiency observed in Figure 4.

## 5 Related Work

Many practical off-policy actor-critic methods, including the SVG algorithm, rely on uniform sampling from the replay buffer. Although importance sampling corrections have been proposed in model-free reinforcement learning (Hallak and Mannor, 2017; Zhang et al., 2020), they are often prone to issues such as high variance in the likelihood ratios (Liu et al., 2019).

In MBRL, MBPO (Janner et al., 2019) introduced DR, an extension of Dyna-style rollout which generates one-step transitions used along with real samples (Sutton, 1990). Although MBPO achieves high sample efficiency, it suffers from extensive runtime requirements. In contrast, while MBPO uses the model solely for generating virtual data, the SVG method leverages the differentiability of the model directly during policy optimization, enabling more efficient learning.

PILCO (Deisenroth and Rasmussen, 2011), which employs Gaussian processes (GPs) as its model, demonstrates very high sample efficiency by performing policy optimization using SVG on the analytical distribution of the cumulative reward. However, when using sample-based predictions instead of assuming an analytical distribution, gradient estimation becomes unstable—especially for long prediction horizons—as errors accumulate over time (Parmas et al., 2018). This issue is particularly pronounced in tasks where accurate long-term predictions are crucial.

Furthermore, as discussed in Section 3, SVG methods have not adequately addressed the issue of distribution shift in the objective function. Specifically, Feinberg et al. (2018) explored the benefits of model-based value estimation and derived conditions for achieving an on-policy stationary distribution; however, the utility of combining these conditions with SVG for policy optimization remained unclear. In this study, we introduce a novel approach that employs the MVE estimator for both actor and critic learning, thereby resolving the distribution shift issue. This constitutes the key novelty of our work.

Finally, while it is possible to use importance sampling (IS) to address distribution shift—as is done in the model-free reinforcement learning literature—our study focuses on achieving practical performance improvements. We demonstrate that simply adding DR leads to performance gains without significantly increasing runtime. This simple yet effective approach is highly beneficial for developing RL technologies where sample efficiency is critical.

## 6 Conclusion

In this study, we introduced DHMBPO, a novel MBRL algorithm that integrates two distinct model rollouts —a long-horizon DR and a short-horizon TR— to address two critical trade-offs: (i) state distribution shift versus model bias, and (ii) value and value gradient accuracy versus policy gradient instability. Long DR resamples from the model to better approximate the on-policy state distribution, while short TR leverages differentiable transitions to provide accurate on-policy value estimation with stable gradient updates, thereby reducing the number of required updates and overall runtime. By assigning different horizon lengths to DR and TR, we preserve sample efficiency without incurring excessive model bias or increased runtime.

Experimental results on multiple continuous-control tasks indicate that DHMBPO consistently achieves faster learning and shorter runtime compared to existing MBRL baselines. We observe that the synergy of a long DR with a short TR enables early improvements in value estimation, which accumulate over successive policy updates. Although the method provides a more balanced solution to the model bias versus off-policy data challenge, exploration remains an open issue. For instance, certain tasks such as `finger-spin` (see Figure 8 in Appendix C) can still lead to local optima. Nonetheless, the relatively short runtime of DHMBPO facilitates iterative refinement and experimentation, offering a cost-effective path toward practical performance gains in reinforcement learning applications where sampling cost is critical.

## Acknowledgement

PP was supported by JSPS KAKENHI Grant Number JP22H04998.

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

# A  Implementation Detail

The neural network architectures for the dynamics and reward model, as well as for the actor and critics, are described in Section A.1. Regarding the dynamics and reward model, Section A.2 explains how the model is used for prediction, while Section A.3 details its training. Hyperparameters are provided in Section A.4. Finally, Section A.5 describes techniques for achieving stable critic predictions.

## A.1  Neural network models

### A.1.1  Dynamics and reward model

Following Janner et al. (2019), we employ a bootstrap ensemble of $M = 8$ models,

$$\{p_{\psi_1}, p_{\psi_2}, \cdots, p_{\psi_M}\}.$$

We refer to them as deep ensemble (DE) models. As discussed in Section A.3, each ensemble member is trained on (virtually) independent bootstrapped datasets. The ensemble models capture aleatoric noise by fitting independent multivariate Gaussian distributions.

Concretely, the $m$-th model $p_{\psi_m}(s, a)$ takes a state-action pair $x = (s, a)$ as input and outputs the mean vector $\mu_m \in \mathbb{R}^{S+1}$ of a multivariate Gaussian distribution. Here, $S$ is the dimension of the continuous state space, and we jointly predict the next state ($S$ dimensions) plus the reward (1 dimension), hence the total output dimension $S + 1$. For the covariance, we assume homoscedastic noise, meaning that each model's diagonal covariance elements $\sigma_m^2 \in \mathbb{R}^{S+1}$ are constant with respect to the input.

We use Layer Normalization (Ba et al., 2016) and Dropout (Srivastava et al., 2014) for regularization, and apply SiLU (Elfwing et al., 2018) as the activation function.

Below is a PyTorch-like summary of our network architecture:

```
(model): Sequential(
    (0): EnsembleLinearLayer(
        8, in_size=S+A, out_size=256, decay=0.00025)
    (1): Dropout(p=0.0075)
    (2): SiLU()
    (3): LayerNorm((256,))
    (4): EnsembleLinearLayer(
        8, in_size=256, out_size=256, decay=0.0005)
    (5): Dropout(p=0.005)
    (6): SiLU()
    (7): LayerNorm((256,))
    (8): EnsembleLinearLayer(
        8, in_size=256, out_size=256, decay=0.00075)
    (9): Dropout(p=0.0025)
    (10): SiLU()
    (11): LayerNorm((256,))
    (12): EnsembleLinearLayer(
        8, in_size=256, out_size=S+1, decay=0.001)
  )
)
```
"EnsembleLinearLayer" denotes the layer which calculates the feed forward pass in parallel.

### A.1.2 Actor and Critics

We also use ensemble of MLP as critic networks used for randomized ensemble double Q-learning, or RED-Q (Chen et al., 2020). We summarize the neural network architecture using Pytorch-like notation:

for model,

```
(critics): Sequential(
    (0): EnsembleLinearLayer(
        5, in_size=S+A, out_size=512, decay=0.0)
    (1) Dropout(p=0.0001)
    (2): SiLU()
    (3) LayerNorm((256,))
    (4): EnsembleLinearLayer(
        5, in_size=512, out_size=512, decay=0.0)
    (5) Dropout(p=0.0001)
    (6): SiLU()
    (7) LayerNorm((256,))
    (8): EnsembleLinearLayer(
        5, in_size=512, out_size=512, decay=0.0)
    (9) Dropout(p=0.0001)
    (10): SiLU()
    (11) LayerNorm((256,))
    (12): EnsembleLinearLayer(
        5, in_size=512, out_size=1, decay=0.0)
  )
)
```
for critic networks, and

```
(actor): Sequential(
    (0): Linear(in_features=S, out_features=512)
    (1): SiLU()
    (2): Linear(in_features=512, out_features=512)
    (3): SiLU()
    (4): Linear(in_features=512, out_features=512)
    (5): SiLU()
    (6): Linear(in_features=512, out_features=2A)
  )
)
```

for actor networks, respectively, where $S$ is the dimension of state space, and $A$ is the dimension of action space. The total number of weight parameters, for example on `Humanoid` task from GYM suite is about 2M, where S=45 and A=17.

## A.2   Model prediction

We propose to normalize the signals targeted for regression (see Section A.3 for more detail). Given a variable $z$, we denote its normalized version by $\bar{z}$. We perform normalization by the empirical mean $\mu_z$ and standard deviation $\sigma_z$ statistics of the variable with the equation $\bar{z} := \frac{z - \mu_z}{\sigma_z}$. Similarly, we can perform denormalization by $z = \bar{z}\sigma_z + \mu_z$. We denote the state vectors and the next-state vectors as $s, s' \in \mathbf{R}^{d_S}$ respectively, and the $i$-th element of the displacement between them as $\Delta_i := s'_i - s_i$. Our dynamics models, $T_\psi$, are trained to predict the displacements in states from one step to the next, thus we normalize the outputs based on their used targets. Let $\mu_{\Delta_i}$ and $\sigma_{\Delta_i}$ be the mean and standard deviation of the observed displacements in the data up to now. During training the model, the $i$-th element of the model's target signal for the $n$-th data point is defined as $\bar{t}_{ni} := \frac{\Delta_{ni} - \mu_{\Delta_i}}{\sigma_{\Delta_i}}$. We summarize the different normalizations we perform: all inputs to models are normalized; the rewards are normalized during training the reward model, but the predicted rewards are denormalized during policy learning; the actions are never normalized (as we choose their scale to be reasonable); the transition model targets are normalized during model training and also during the actor training rollouts.

The rollout predictions, in particular, require care in the implementation. Suppose the normalized state is $\bar{s}$ and the transition model has predicted a change $\bar{\Delta}$. We can add the normalized displacement to the normalized state by first denormalizing, then adding them, then normalizing them again. This leads to the process: (i) $s = \mu_s + \sigma_s\bar{s}$, (ii) $\Delta = \mu_\Delta + \sigma_\Delta\bar{\Delta}$, (iii) $s' = s + \Delta = \mu_s + \sigma_s\bar{s} + \mu_\Delta + \sigma_\Delta\bar{\Delta}$, (iv) $\bar{s}' = \frac{s' - \mu_s}{\sigma_s} = \bar{s} + \frac{\mu_\Delta + \sigma_\Delta\bar{\Delta}}{\sigma_s}$. We thus see that to make correct predictions in the normalized space using our displacement model, we need to denormalize the displacements, then normalize them using the normalization scale of the states, $s$, and add the displacement to the normalized state, $\bar{s}$.

Note that although a similar normalization was performed in SAC-SVG(H), the output was normalized using the sample mean and standard deviation of the state, just like the input.

When predicting next-state and reward during rollout, we adopt the TS-1 algorithm proposed in Chua et al. (2018), which randomly selects an ensemble member for each input and timestep in order to propagate uncertainty through bootstrapping.

## A.3   Model Learning

### A.3.1   The objective function

The objective of the model learning is minimizing the sum of the negative log-likelihood:

$$\mathcal{L}(\psi) := \frac{1}{MND} \sum_{m=1}^{M} \sum_{n=1}^{N} w_{mn}\mathcal{L}_{mn}$$

where

$$\mathcal{L}_{mn} := \sum_{d=1}^{D} \left\{ \log \sigma_{md} + \frac{1}{2} \left( \frac{\bar{t}_{nd} - \mu_{mnd}}{\sigma_{md}} \right)^2 \right\}.$$

$\mu_{mnd} \coloneqq \mu_{\psi_m}^{(d)}(x_n)$ represents the $d$th output of the $m$th neural network $\mu_{\psi_m}$ for the $n$th input $x_n \coloneqq (s_n, a_n)$, and $\sigma_{md}$ denotes the noise regarding the $d$th dimension output of the $m$th ensemble member $\sigma_m$.

$w_{mn} \in [0, \infty]$ is a weight for $n$th sample allocated to the $m$th model, in order to make the dataset virtually independent bootstrapped one. The value is randomly sampled from the exponential distribution $f(w|1)$ with the parameter $\lambda = 1$ and fixed, thus the effective number of data used for each model's learning is equivalent on average.

### A.3.2 Early stopping

When $x_n$ is the input, let $\mu_{mn}$ and $\sigma_m$ represent the mean and scale parameter, of the $m$th model's predicted Gaussian distribution. We aggregate the predictions across all ensemble members, and define the prediction of the DE model as a Gaussian distribution with mean parameters for the $d$th dimension when the input is $x_n$ as $\tilde{\mu}_{nd} \coloneqq \frac{1}{M} \sum_{m=1}^{M} \mu_{mnd}$ and the scale parameter as $\tilde{\sigma}_{nd}^2 \coloneqq s_{nd}^2 + \frac{1}{M} \sum_{m=1}^{M} \sigma_{md}^2$, where $s_{nd}^2$ is the sample variance among the ensemble of predictions $\{\mu_{mnd}\}_{m=1}^{M}$. This approach is often also called moment matching.

Then, we define the evaluation score $S_n$ for the $n$th data as:

$$S_n \coloneqq \frac{1}{2} \sum_{d=1}^{D} \left[ \frac{(\tilde{\mu}_{nd} - t_{nd})^2}{\tilde{\sigma}_{nd}^2} + \log \tilde{\sigma}_{nd}^2 \right]$$

under the assumption that the predicted distribution follows a Gaussian distribution with a diagonal co-variance matrix. After each epoch of model learning, we evaluate the score for each validation data point and compare the distribution of these scores based on the new parameters of the model with those from the previous parameters. If the distribution of the new scores shows significant improvement according to a Z-test with a significance level of 0.1, we continue training the model.

Our stopping strategy is the same as in the MBPO implementation: if the new model does not show improvement over a fixed number of epochs, the training is stopped. We set the threshold as $b \log d_s$, where $b$ is a scalar-valued hyper-parameter, representing the base number of epochs, and $d_s$ is the dimension of the state space.

### A.4 Hyper-parameters

We summarize in Table 2 hyper-parameters shared for all experiments in Section 4, for DHMBPO and SAC. Since the DHMBPO algorithm shares common modules with the SAC implementation, such as the architecture of the critic network, the hyperparameters and their values are also identical. Exceptions to this are explicitly indicated in parentheses.

### A.5 Bounding target critic's output

In this study, we use the MVE estimator for $Q$-function, eq. (9) as the target values in critic learning. However, due to the recursive dynamics model predictions, the model errors may accumulate or value estimation errors may increase (Buckman et al., 2018). In order to avoid this risk, we bound the values predicted by the target critics to stabilize critic learning.

Since the $Q$-value is within $\frac{1}{1-\gamma}$ times the upper and lower bounds of the reward value (*Proof.* $\sum_{h=0}^{\infty} \gamma^h r_h \leq \sum_{h=0}^{\infty} \gamma^h r_{\max} = \frac{1}{1-\gamma} r_{\max}$ and similar for the lower bound), we approximate these upper and lower reward bounds. The procedure for estimation of bounds on $Q^{\pi_\theta}$ is as follows: (1) Perform training rollouts. (2) Set the 1-st and 99-th percentiles of rewards from the rollouts as $r_l$ and $r_u$, respectively. (3) $Q_l \coloneqq \frac{r_l}{1-\gamma}$ and $Q_u \coloneqq \frac{r_u}{1-\gamma}$. (4) When calculating the bounds for the first time, set $Q_l$ and $Q_u$ as the initial values, and subsequently update them by taking exponential moving averages with a hyperparameter $\eta$. Then we clip the target critic networks' outputs in eq. (9) by $Q_l$ and $Q_u$. Since the process is simple, its computation cost is negligible.

Table 2: Hyperparameters commonly set for DHMBPO and SAC across all experiments. The first half of the table presents the hyperparameters shared between DHMBPO and SAC.

| Hyper-parameter | Value |
|---|---|
| Discount factor | 0.995 |
| Seed steps | 5000 |
| Action repeat | 1 (Gym) |
| | 2 (DMControl) |
| Batch size | 256 |
| Update-to-data ratio | 1 |
| Replay buffer size | 1M |
| Learning rate for the actor, critics and $\alpha$ | $3 \cdot 10^{-4}$ |
| Initial value of $\alpha$ | 0.1 |
| Momentum coefficient $c$ for target critic | 0.995 |
| Ensemble size of critic | 5 |
| Length of DR $D$ | 20 |
| Length of training rollout $T$ | 5 |
| Iteration per DR | 20 |
| Ensemble size of model | 8 |
| Optimizer for training model | AdamW (Loshchilov and Hutter, 2019) |
| Learning rate for model | $1 \cdot 10^{-3}$ |

We do not perform, on the other hand, clipping on the output of the critic used when training actor with eq. (7). This is because if the critic network outputs a value outside the bounds, the gradient to encourage predicting values within that range would vanish because of the clipping.

The motivation for using percentiles in the step (2), instead of using a hard max and min, is based on the observation in our preliminary experiments (Appendix D.7). There are occasional large outliers in the rewards that would make the bounds extremely broad. Ignoring these outliers, but estimating the bounds based on the most of the samples gives more efficient and stable results.

## B Experiment Details

We describe configuration for comparison in Section 4.2. In this study, when the other algorithms' experimental results under identical settings were available, we used those results as part of the results (MBPO and Dreamer v3). Instead of using the official implementation by Haarnoja et al. (2018), we utilized results based on our own implementation of the SAC algorithm. This approach allows for a high degree of implementation commonality between SAC and DHMBPO, except for the components involving the model. Specifically, as described in Appendix A.4, we shared various implementation aspects, including the architecture of the critic network. For the rest of the results, we ran the publicly available source code with its default hyperparameter settings.

The number of samples to measure a test return varies depending on the implementation of the algorithm. Like DHMBPO, SAC-SVG(H) (Amos et al., 2021), RP-PGM (Zhang et al., 2023), SAC (Haarnoja et al., 2018), and TD-MPC2 (Hansen et al., 2024) used 10 test episodes, while as MACURA (Frauenknecht et al., 2024) used 3 test episodes and MBPO (Janner et al., 2019; Pineda et al., 2021) and Dreamer v3 (Hafner et al., 2023) used 1 test episode.

For each GYM tasks, except for DHMBPO, each algorithms' hyper-parameters were tuned per-task. Following (Janner et al., 2019), we used variants for the `Humanoid` and `Ant` environments from GYM, with some state dimensions truncated to lower dimensions.

For DMC tasks, Each of three algorithms' hyper-parameters were set commonly over all DMC tasks.

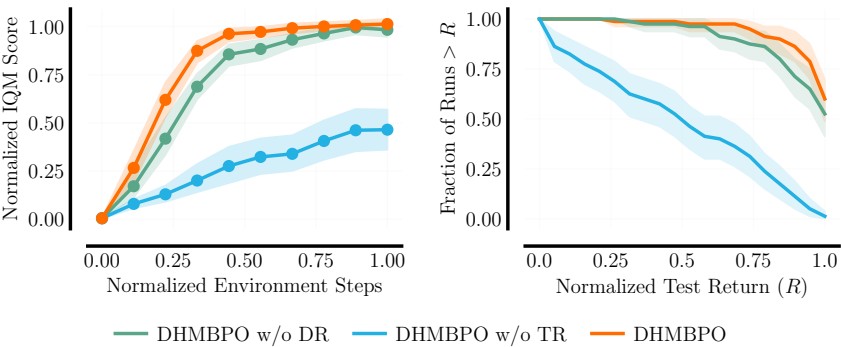

Figure 7: Performance profile (*left*) and score distributions (*right*) of ablated algorithms on 5 GYM tasks and 5 DMC tasks.

## C   Result Details

### C.1   Ablation study

Figure 7 shows performance profile and score distributions of ablated algorithms on 5 GYM tasks and 5 DMC tasks, of which individual plots are shown in Section 4.3. We see that DHMBPO's statistically significant sample efficiency in early phase against the ablated algorithms.

### C.2   Comparison with other algorithms

For DMC tasks, we provide plots for individual sample efficiency curve in Figure 8, plots for runtime in Figure 9 and Table 3 for summarizing their runtimes. For reference, we include horizontal lines indicating the asymptotic performance of SAC. These lines represent the average test return across 5 random seeds, at ten times the maximum environment step budget used for the model-based methods in each task.

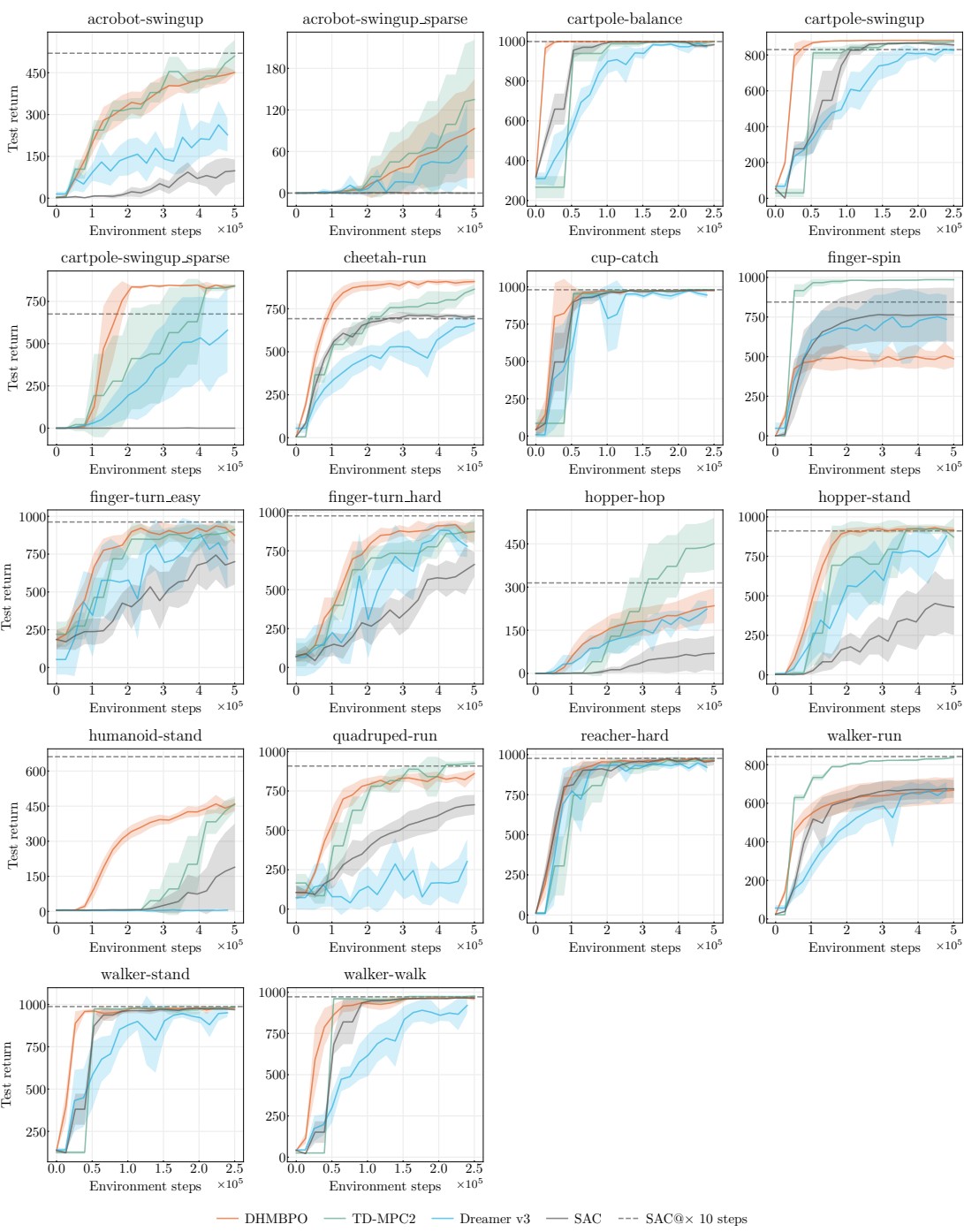

Figure 8: Comparison of sample efficiency with latent model-based methods on DMC tasks. The solid line represents the sample mean over 8 random seeds, and the shaded area indicates the 95% confidence intervals.

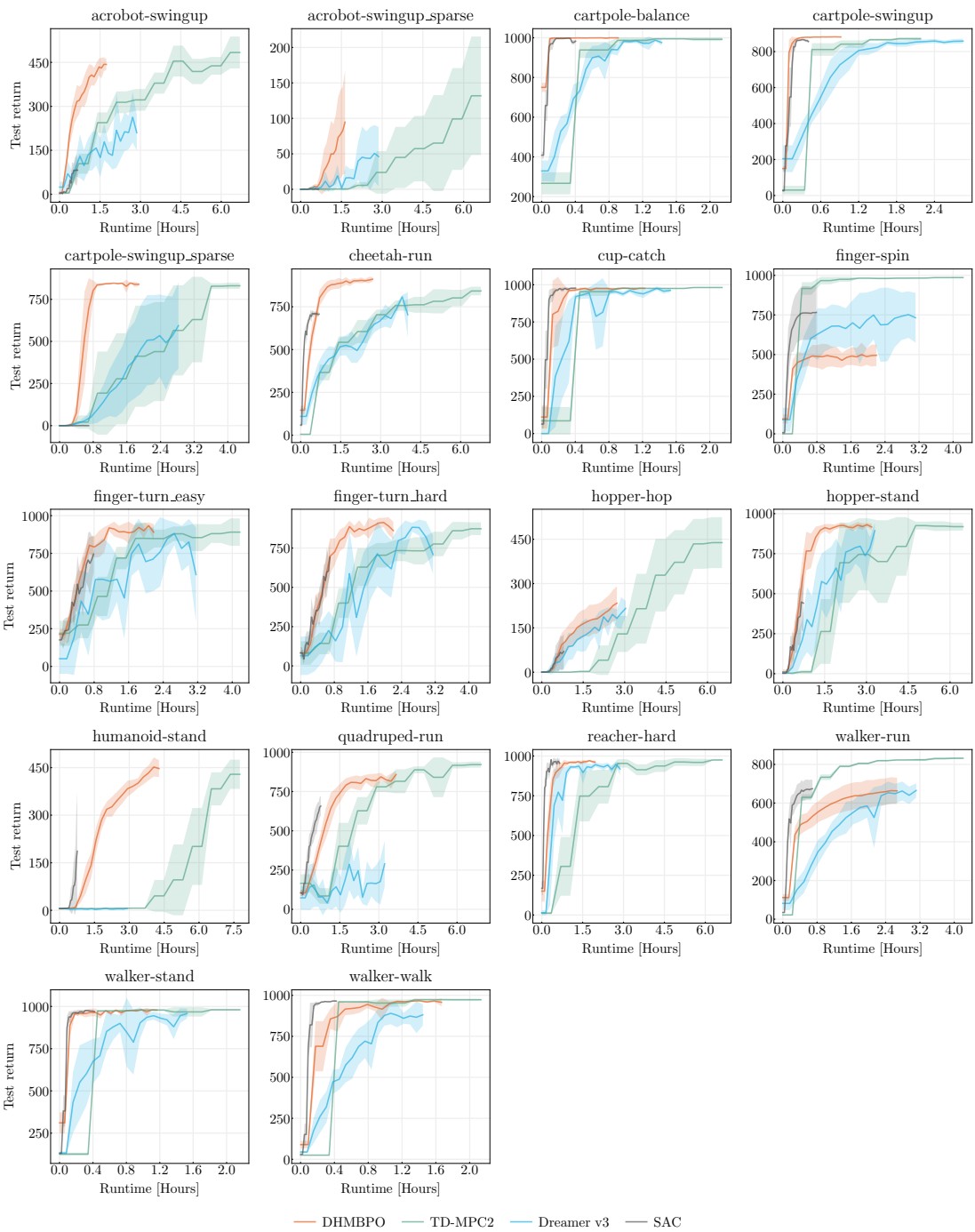

Figure 9: Runtime comparison with latent model-based methods on DMC tasks.

Table 3: Runtimes on DMC tasks until maximum environment step budget (250K or 500K) and its ratio to DHMBPO's mean runtime (in hours).

|  | DHMBPO | Dreamer v3 | TD-MPC2 |
|---|---|---|---|
| acrobot-swingup | 1.7 | 2.9 | 3.8 |
| acrobot-swingup_sparse | 1.6 | 2.9 | 2.6 |
| cartpole-balance | 0.9 | 1.5 | 1.2 |
| cartpole-swingup | 0.9 | 2.9 | 1.9 |
| cartpole-swingup_sparse | 1.9 | 2.9 | 2.3 |
| cheetah-run | 2.7 | 4.1 | 2.8 |
| cup-catch | 1.2 | 1.6 | 1.9 |
| finger-spin | 2.2 | 3.2 | 2.9 |
| finger-turn_easy | 2.2 | 3.2 | 2.7 |
| finger-turn_hard | 2.2 | 3.2 | 2.6 |
| hopper-hop | 2.7 | 3.1 | 3.8 |
| hopper-stand | 3.2 | 3.3 | 2.3 |
| humanoid-stand | 4.3 | 2.9 | 5.0 |
| quadruped-run | 3.7 | 3.2 | 2.7 |
| reacher-hard | 2.0 | 2.9 | 3.7 |
| walker-run | 2.7 | 3.2 | 3.9 |
| walker-stand | 1.2 | 1.6 | 1.8 |
| walker-walk | 1.7 | 1.5 | 3.7 |
| Ratio | 1.0 | 1.3 | 1.3 |

### C.3   Hyper-parameter sensitivity

In Figure 10, we show individual results of experiments in Section 4.4 with respect to sample efficiency and to an Euclidean norm of policy gradient $\theta \in \mathcal{R}^D$: $\sqrt{\sum_i^D \theta_i^2}$.

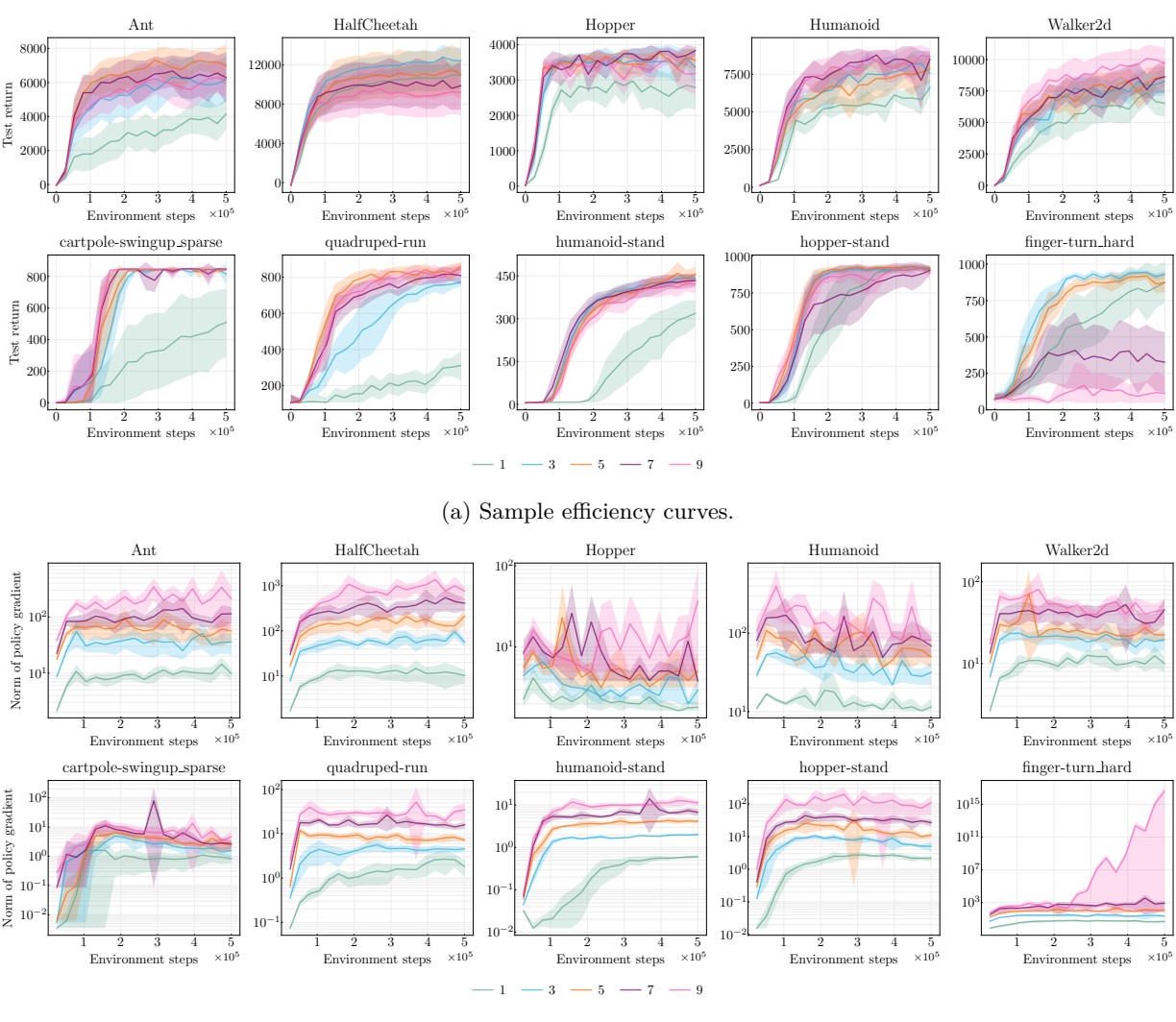

(a) Sample efficiency curves.

(b) Norm of policy gradient.

Figure 10: Development of test return and critic loss over environment steps for different TR lengths

# D   Additional Experiments for Highlighting improvement over DR-based and TR-based algorithms

## D.1   DHMBPO without Training Rollout but with different UTD ratios

We show the results of DHMBPO without TR but with different UTD ratios in Figure 11 for sample efficiency curves, in Figure 13 for performance profiles, in Figure 12 for runtime comparison and in Table 4 for summary of runtimes. We can see that the higher UTD ratio the higher performance. Conversely, DR-based algorithm, or DHMBPO without TR, needs higher UTD ratio in order to attain high sample efficiency, requiring longer runtime as shown in 12. And this results reflect that MBPO (and MACURA) were sample efficient while took so long runtime.

## D.2   Trade-off in Policy Gradient Estimation

We investigate how the horizon length $T$ of the training rollout (TR) affects the bias and variance of value gradient estimates. In principle, increasing $T$ reduces the bias by incorporating longer reward sequences, but it can simultaneously raise the variance of the policy gradient.

**Experimental Setup**

For each benchmark task, we train a DHMBPO agent for 500k environment steps and then fix its policy, critic, dynamics, and reward networks, as well as the replay buffer. We draw $B = 2048$ states $\{s_b\}_{b=1}^{B}$ at random from the replay buffer. For each state $s_b$ and each TR horizon $t \in \{1, 3, 5, 7, 9\}$, we run $R = 2048$ short-horizon rollouts under the learned model and compute the corresponding MVE-based value gradients $\{g_{tbr}\}_{r=1}^{R}$, where $g_{tbr}$ is a normalized vector by the dimension of policy parameter which differs across tasks. We define

$$g_{tb} \coloneqq \frac{1}{R}\sum_{r=1}^{R} g_{tbr}, \quad g_t \coloneqq \frac{1}{B}\sum_{b=1}^{B} g_{tb}.$$

Following the approach in our hyperparameter sensitivity analysis (Appendix C.3), we designate $t = 9 =: T$ as the ground truth horizon:

$$\bar{g} \coloneqq g_T.$$

In other words, we treat the gradient computed at $t = 9$ as a baseline for evaluating shorter horizons.

Table 4: Runtime until 500K environment step. Tha last row shows the ratio of mean runtime to that of the case with UTD = 1.

| The UTD ratio | 1 | 2 | 4 | 8 | 16 |
|---|---|---|---|---|---|
| Ant | 3.8 | 5.1 | 7.9 | 13.7 | 27.0 |
| HalfCheetah | 3.0 | 4.4 | 6.6 | 11.8 | 23.2 |
| Hopper | 3.3 | 4.7 | 7.5 | 12.6 | 26.5 |
| Humanoid | 4.6 | 6.8 | 9.7 | 15.9 | 28.1 |
| Walker2d | 3.6 | 5.2 | 8.7 | 14.5 | 26.7 |
| cartpole-swingup_sparse | 1.7 | 2.3 | 3.7 | 6.3 | 12.2 |
| finger-turn_hard | 2.3 | 2.3 | 3.5 | 6.3 | 12.9 |
| hopper-stand | 2.1 | 2.6 | 4.5 | 6.8 | 11.2 |
| humanoid-stand | 3.7 | 4.5 | 5.8 | 8.0 | 14.6 |
| quadruped-run | 3.8 | 4.2 | 5.4 | 7.7 | 11.9 |
| Ratio | 1.0 | 1.3 | 2.0 | 3.3 | 6.1 |

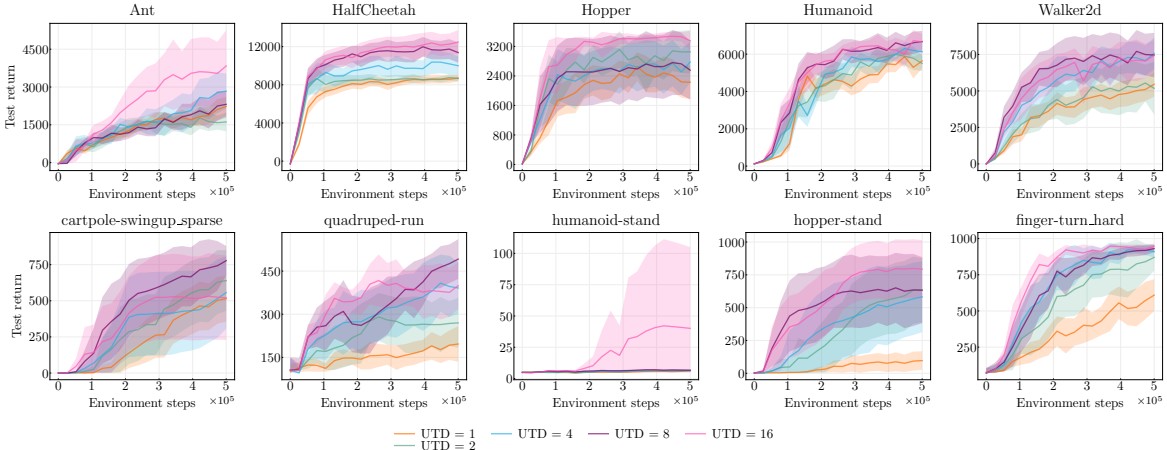

Figure 11: Sample efficiency curves of DHMBPO without TR (corresponding to MBPO) but with different UTD ratios.

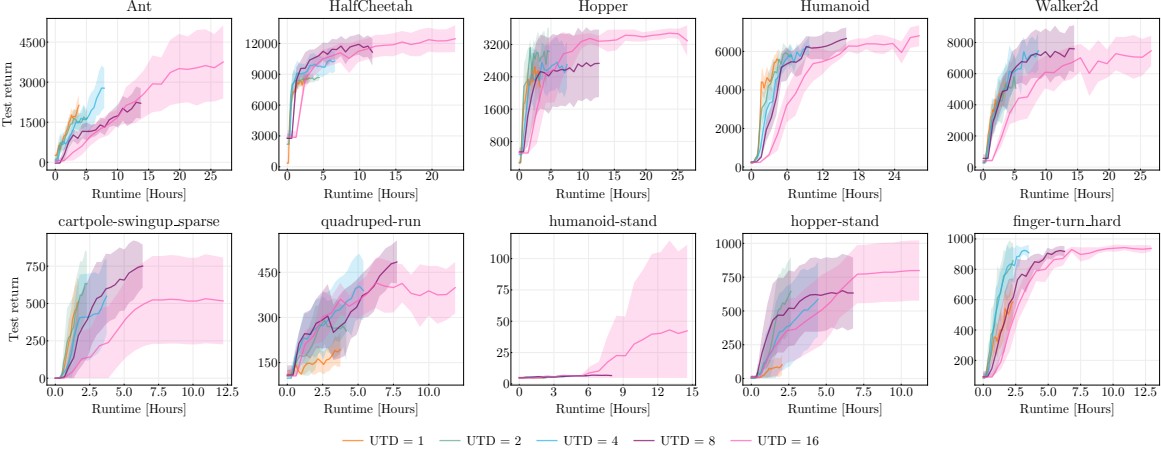

Figure 12: Runtime comparison of the DHMBPO variants without TR but with different UTD ratios.

To quantify bias, we measure the mean squared difference between each sample-averaged gradient $g_{tb}$ and $\bar{g}$, averaged over all $b$:

$$\text{Bias}_t := \frac{1}{B} \sum_{b=1}^{B} \left\| g_{tb} - \bar{g} \right\|^2.$$

In addition, we compute the *standard error of the mean* (SEM) to capture the variability of the gradient across states in the replay buffer:

$$\text{SEM}_t := \sqrt{\frac{\sum_{b=1}^{B} \left\| g_{tb} - g_t \right\|^2}{B(B-1)}},$$

where $\| \cdot \|$ denotes the Euclidean norm over the policy parameter dimension (i.e., the gradient dimension). Intuitively, $\text{Bias}_t$ reflects how far the sample-averaged gradient at horizon $t$ is from the ground truth at $t = 9$, while $\text{SEM}_t$ indicates how much variability remains across different states.

**Experimental Results**

Figure 14 presents results using $T = 5, 7$, and $9$ as the ground-truth horizon. In all tasks except `finger-turn_hard`, we see that as $t$ increases from 1 to 9, the bias steadily declines while the SEM

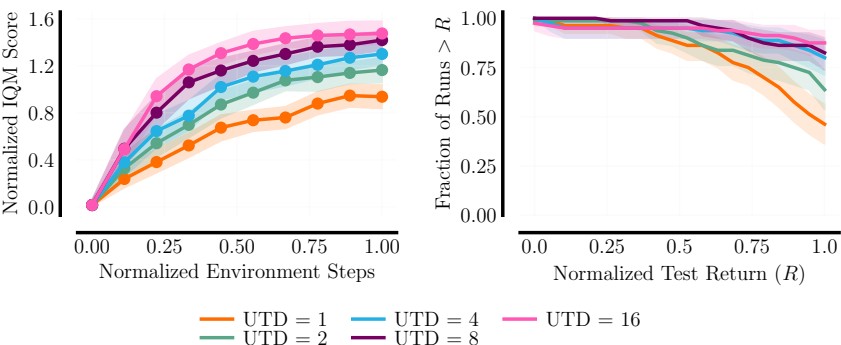

Figure 13: Performance profiles and score distributions of DHMBPO without TR (corresponding to MBPO) but with different UTD ratios. The baseline here is the variant with UTD = 1.

grows—revealing a clear trade-off between bias and variance in the policy gradient. By contrast, in `finger-turn_hard`, the bias when T=9 did not show clear trend to decrease, suggesting the batch size of 2048 was not enough for $g_{T=9}$ to be as ground truth, and that setting $t = 5$ does not necessarily prevent gradient explosion in certain challenging domains.

Notably, all these estimates rely on a replay buffer of 2048 real-environment states, while the multiple trajectories at each state are generated by the model, although the model was trained on after 500k samples. Therefore, this method can be carried out *without* any additional real-environment interaction and used to set relatively stable length of TR horizon, as demonstrated in unclear trend of the bias in `finger-turn_hard`.

## D.3 Application of DR to SAC-SVG(H) Implementation

In order to see the effect of DR on an existing implementation, we added DR procedure into the official code of SAC-SVG(H) algorithm (Amos et al., 2021), which employs the SVG method with a single GRU (Cho, 2014) as the recurrent neural network and one-step TD errors for training the critic networks. We refer to this variant as DHSAC-SVG(H) and evaluated it on five GYM tasks with varying DR lengths, repeating the execution with different 5 seeds. As shown in Fig. 15, the variants with longer DRs exhibited either poor learning performance or a drop in performance after a certain point. We investigated the training metrics and found that the critic losses started to explode as the DR length increased, as shown in Fig. 16. Here, the solid line is mean $\mu$ and shaded area shows the 95% CIs in log-scale: $\mu(1 \pm \exp(\log \text{SEM} \times 1.96))$ where $\log \text{SEM}$ is defined as $\log \text{SEM} := \sqrt{\frac{\sum_{s=1}^{S}(l_s - \mu)^2}{S(S-1)}}$ and $l_s$ is critic loss with $s$-th seed.

One potential cause of these results is the deterministic nature of the model used in the SAC-SVG(H) algorithm. Predictions generated by a fixed deterministic model are likely to introduce consistent biases. In contrast, the DE model approach adopted by DHMBPO and MBPO uses multiple ensemble members, each potentially having similar biases. However, the manner in which these biases accumulate is random across ensemble members. During predictions, an ensemble member is randomly selected at each timestep, making the bias accumulation in the predicted states also random. As a result, within a single trajectory, these biases may cancel out to some extent, leading to an overall smaller average bias, albeit with larger variance.

Policy evaluation and improvement based on the higher-variance predictions of a DE model may slow down learning, but they avoid the risk of extreme predictions caused by deterministic models with large biases. On the other hand, policy evaluation and optimization using predictions with significant bias from deterministic models carry the risk of focusing on extreme values, potentially destabilizing the training process.

### D.4 Comparison of Model Learning Performance

We also conducted a quantitative analysis to compare the predictive performance of models used in DHMBPO, SAC-SVG(H), and MBPO (Pineda et al., 2021), using common training and evaluation datasets.

**Data Generation Procedure**   For each GYM task, we ran the DHMBPO algorithm for up to 500K steps. From the replay buffer, we created three input-output datasets containing data collected up to 5K, 20K, and 50K steps, respectively. Additionally, we created another dataset from all data collected between 495K and 500K steps. Each dataset was split into training (80%) and validation (20%) subsets, while the 495K–500K dataset was exclusively used for evaluation.

**Evaluation Methodology**   For each of the three models, we trained them using the respective training datasets and measured Root Mean Squared Error (RMSE) on the training, validation, and evaluation datasets at each epoch. This procedure was repeated for three random seeds.

**Results**   The results of the experiments are plotted separately for each dataset size, comparing training data versus validation data and training data versus evaluation data, as shown in Figs. 17– 19. Additionally, the results after 100K epochs are summarized in Table 5. The shaded areas in the plots and the $\pm$ values in the table both represent the standard error over three seeds.

Table 5: Summary of RMSE values after 100K epochs. "Size" denotes the size of training data.

| Metric | Task | Method Size | DHMBPO | MBPO | SAC-SVG(H) |
|---|---|---|---|---|---|
| Training RMSE | Ant | 5000 | 0.29 ± 0.0 | 1.86 ± 0.02 | 1.1 ± 0.01 |
| | | 20000 | 0.48 ± 0.0 | 1.21 ± 0.01 | 0.67 ± 0.0 |
| | | 50000 | 0.5 ± 0.0 | 0.79 ± 0.02 | 0.59 ± 0.0 |
| | HalfCheetah | 5000 | 0.18 ± 0.0 | 1.46 ± 0.01 | 0.7 ± 0.01 |
| | | 20000 | 0.35 ± 0.0 | 0.73 ± 0.01 | 0.57 ± 0.0 |
| | | 50000 | 0.49 ± 0.0 | 1.58 ± 0.08 | 0.76 ± 0.0 |
| | Hopper | 5000 | 0.01 ± 0.0 | 0.06 ± 0.0 | 0.11 ± 0.0 |
| | | 20000 | 0.02 ± 0.0 | 0.09 ± 0.0 | 0.18 ± 0.0 |
| | | 50000 | 0.02 ± 0.0 | 0.04 ± 0.0 | 0.21 ± 0.0 |
| | Humanoid | 5000 | 0.44 ± 0.0 | 2.02 ± 0.0 | 2.99 ± 0.02 |
| | | 20000 | 1.18 ± 0.0 | 1.99 ± 0.0 | 1.65 ± 0.0 |
| | | 50000 | 1.55 ± 0.0 | 1.77 ± 0.01 | 1.69 ± 0.0 |
| | Walker2d | 5000 | 0.16 ± 0.0 | 1.74 ± 0.08 | 2.88 ± 0.01 |
| | | 20000 | 0.36 ± 0.0 | 1.1 ± 0.01 | 1.48 ± 0.03 |
| | | 50000 | 0.43 ± 0.0 | 0.72 ± 0.02 | 0.92 ± 0.0 |
| Validation RMSE | Ant | 5000 | 6.44 ± 0.03 | 4.29 ± 0.12 | 8.59 ± 0.07 |
| | | 20000 | 1.51 ± 0.02 | 1.62 ± 0.02 | 3.64 ± 0.03 |
| | | 50000 | 0.93 ± 0.01 | 1.07 ± 0.01 | 1.43 ± 0.01 |
| | HalfCheetah | 5000 | 3.72 ± 0.01 | 3.82 ± 0.03 | 8.78 ± 0.01 |
| | | 20000 | 1.99 ± 0.02 | 1.87 ± 0.03 | 3.16 ± 0.05 |
| | | 50000 | 1.61 ± 0.03 | 1.97 ± 0.11 | 3.9 ± 0.14 |
| | Hopper | 5000 | 0.19 ± 0.01 | 0.16 ± 0.01 | 0.25 ± 0.0 |
| | | 20000 | 0.25 ± 0.02 | 0.2 ± 0.0 | 0.95 ± 0.03 |
| | | 50000 | 0.13 ± 0.01 | 0.1 ± 0.01 | 1.15 ± 0.02 |
| | Humanoid | 5000 | 5.14 ± 0.06 | 5.09 ± 0.02 | 7.9 ± 0.1 |
| | | 20000 | 3.25 ± 0.04 | 3.49 ± 0.02 | 9.25 ± 0.08 |
| | | 50000 | 2.66 ± 0.01 | 2.52 ± 0.02 | 6.51 ± 0.0 |
| | Walker2d | 5000 | 4.08 ± 0.06 | 4.54 ± 0.16 | 6.04 ± 0.04 |
| | | 20000 | 2.69 ± 0.02 | 2.79 ± 0.08 | 7.22 ± 0.06 |
| | | 50000 | 1.46 ± 0.02 | 1.54 ± 0.03 | 2.49 ± 0.02 |
| Evaluation RMSE | Ant | 5000 | 8.61 ± 0.21 | 6.45 ± 0.18 | 10.9 ± 0.22 |
| | | 20000 | 2.49 ± 0.06 | 2.05 ± 0.05 | 6.29 ± 0.11 |
| | | 50000 | 1.13 ± 0.01 | 1.25 ± 0.01 | 2.61 ± 0.02 |
| | HalfCheetah | 5000 | 21.11 ± 0.24 | 274.94 ± 51.26 | 25.31 ± 1.15 |
| | | 20000 | 12.76 ± 0.19 | 18.12 ± 1.23 | 16.56 ± 0.35 |
| | | 50000 | 5.89 ± 0.06 | 27.91 ± 4.62 | 9.51 ± 0.23 |
| | Hopper | 5000 | 1.97 ± 0.01 | 20.67 ± 1.34 | 2.05 ± 0.03 |
| | | 20000 | 0.99 ± 0.04 | 1.47 ± 0.19 | 1.61 ± 0.13 |
| | | 50000 | 0.4 ± 0.0 | 0.62 ± 0.16 | 0.88 ± 0.01 |
| | Humanoid | 5000 | 12.91 ± 0.18 | 14.25 ± 0.54 | 17.22 ± 0.69 |
| | | 20000 | 12.64 ± 0.21 | 14.63 ± 0.19 | 13.24 ± 0.46 |
| | | 50000 | 11.47 ± 0.18 | 11.46 ± 0.17 | 12.86 ± 0.14 |
| | Walker2d | 5000 | 11.27 ± 0.28 | 251.58 ± 54.54 | 13.91 ± 0.59 |
| | | 20000 | 11.37 ± 0.49 | 16.42 ± 2.58 | 12.38 ± 0.76 |
| | | 50000 | 8.89 ± 0.18 | 22.69 ± 10.32 | 9.72 ± 0.3 |

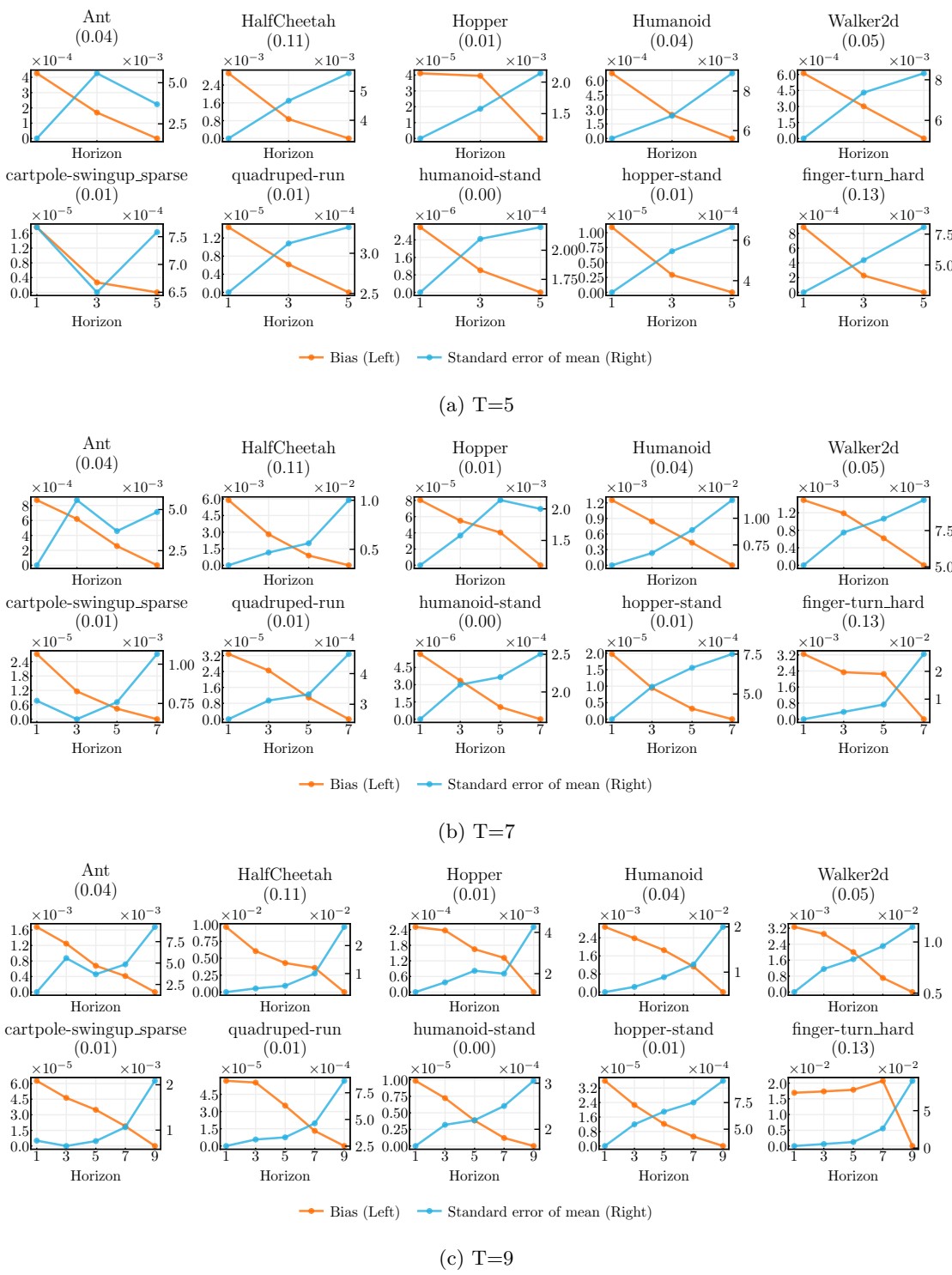

Figure 14: Comparison of different ground-truth horizons. Each plot shows how the bias (left vertical axis) and SEM (right vertical axis) of the value gradient vary with the TR horizon $t$, using either $T = 5, 7,$ or $9$ as the "ground truth" baseline ($\bar{g}$). Across all three choices of ground-truth horizon, we observe a consistent trend: the bias decreases with longer $t$, whereas the SEM increases.

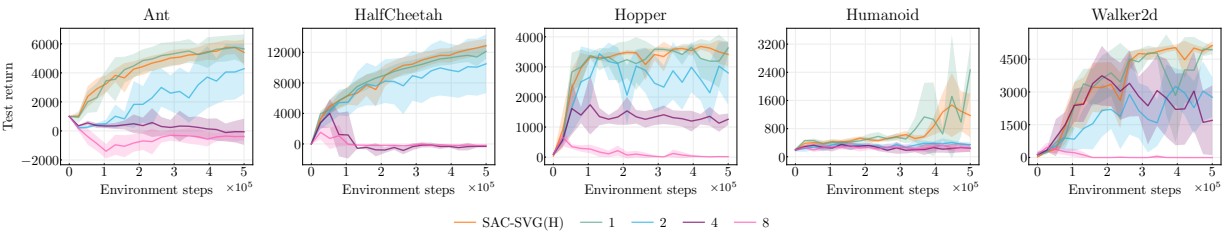

Figure 15: Sample efficiency curve of DHSAC-SVG(H).

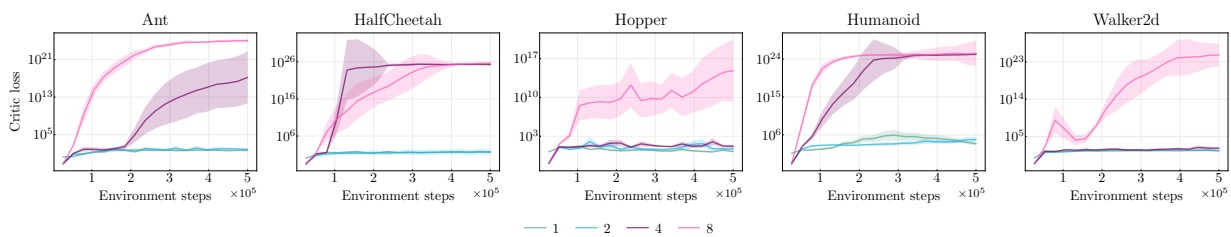

Figure 16: Development of critic loss over the number of environment steps of DHSAC-SVG(H). Solid lines is mean and shade are is for 95% CIs in log-scale. See the text in Appendix D.3 for the detail of the CIs.

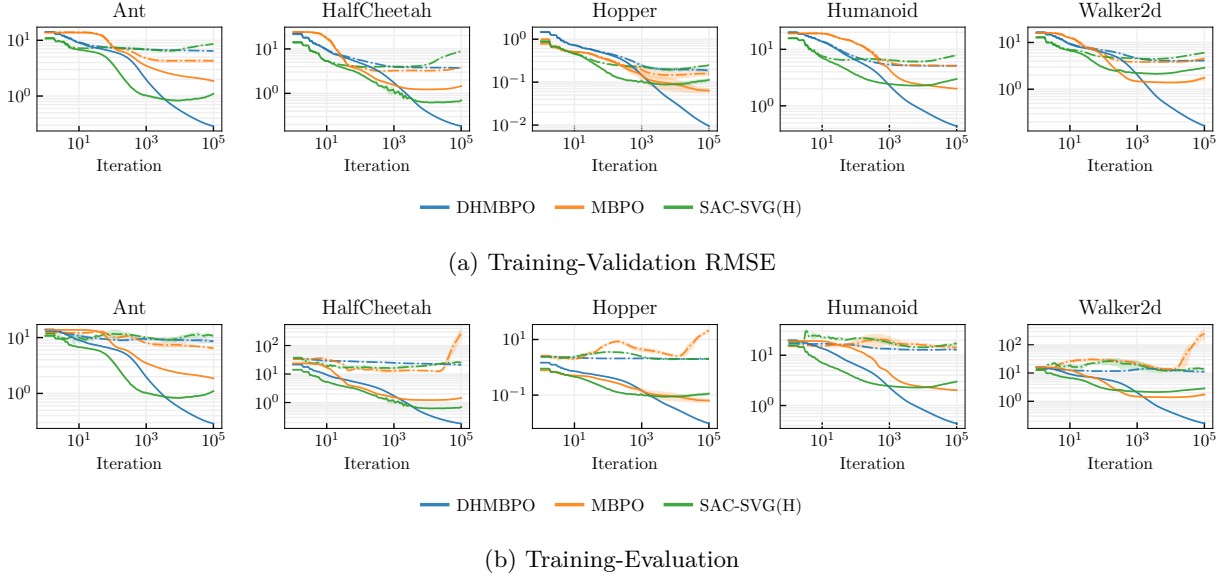

Figure 17: RMSE of model trained on 5K environment steps dataset. Solid line is for training RMSE and dashdot line is for validation RMSE or evaluation RMSE

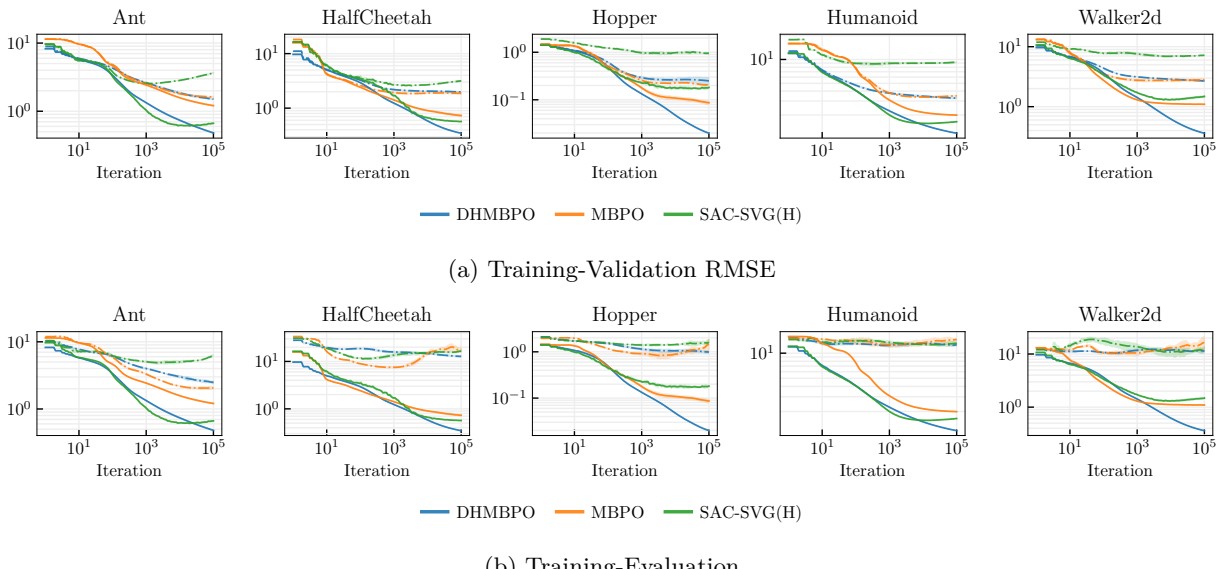

(a) Training-Validation RMSE

(b) Training-Evaluation

Figure 18: RMSE of model trained on 20K environment steps dataset. Solid line is for training RMSE and dashdot line is for validation RMSE or evaluation RMSE

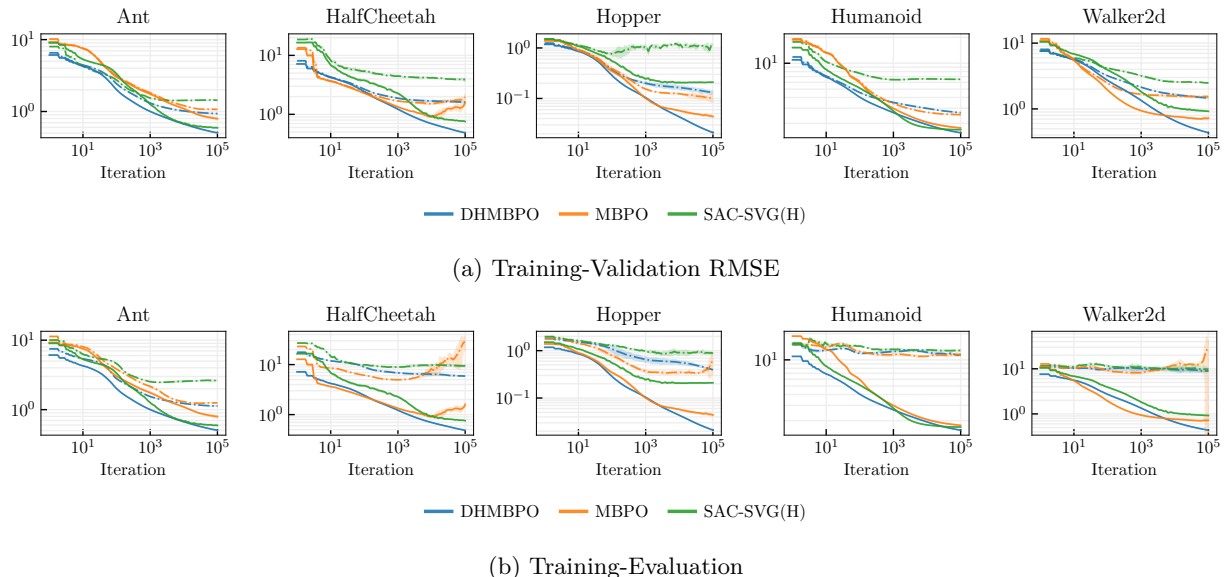

(a) Training-Validation RMSE

(b) Training-Evaluation

Figure 19: RMSE of model trained on 50K environment steps dataset. Solid line is for training RMSE and dashdot line is for validation RMSE or evaluation RMSE

### D.5 Different UTD ratio

We executed DHMBPO with the different UTD ratios, shown in Figure 20. All value for each metrics are normalized by the value for the UTD ratio = 1.

### D.6 Ablation study of model architecture on online RL performance

We additionally conducted ablation study on the network architecture of the dynamics and reward models to Figures 21 and 22. Specifically, we evaluated three variants: (DHMBPO w/o LayerNorm) a model without LayerNorm, (DHMBPO w/o Dropout) a model without Dropout, and (DHMBPO w/ ReLU) a model in which all activation functions were changed from SiLU to ReLU.

From the results in Figure 21, we observed that removing LayerNorm caused the critic loss to diverge and led to a decline in policy performance. For the other two variants, the critic loss did not diverge; however, as shown in the aggregation metrics in Figure 22, their performance significantly deteriorated.

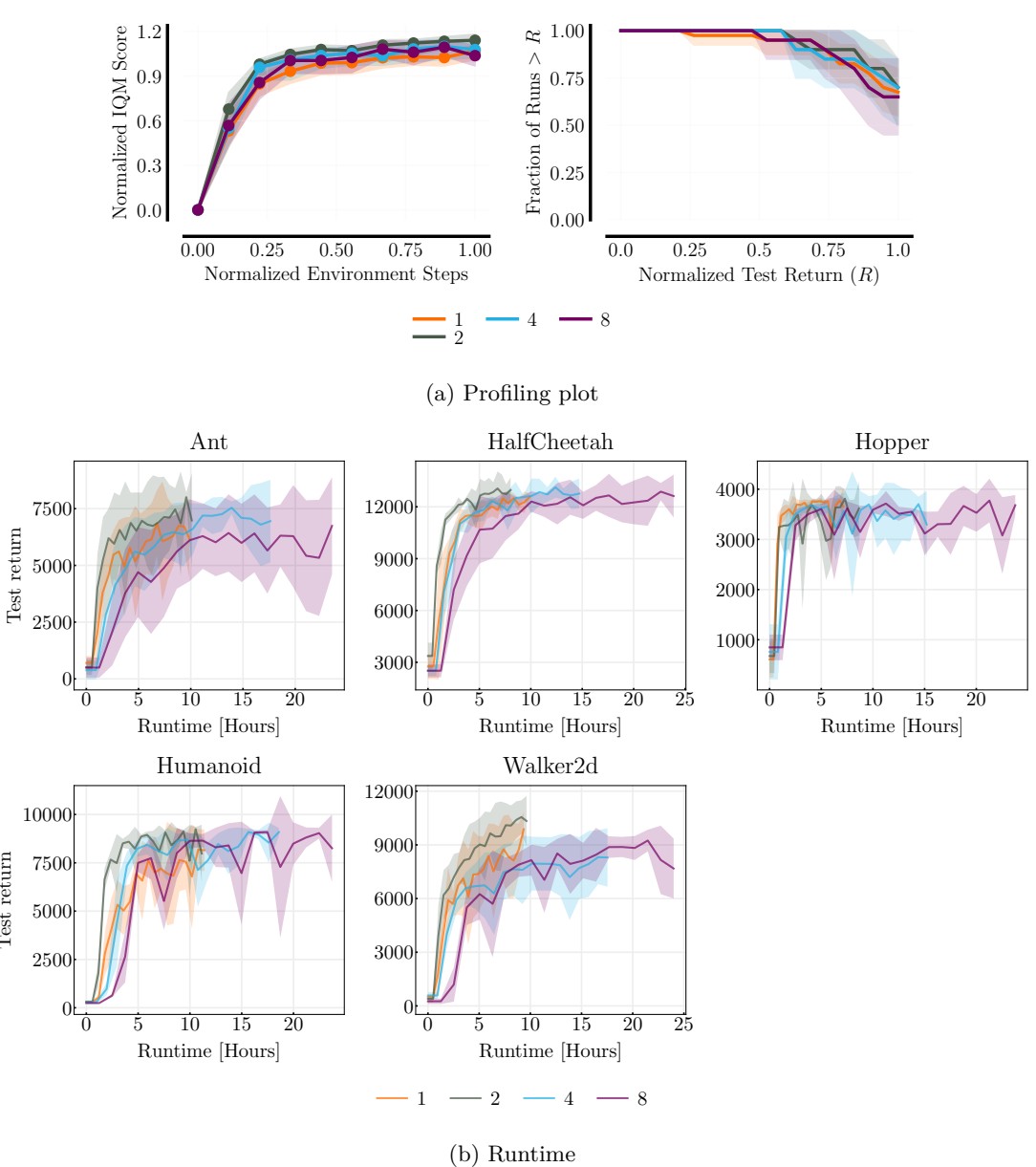

(a) Profiling plot

(b) Runtime

Figure 20: Results on GYM tasks of DHMBPO with different UTD ratios. Legends are for the UTD ratios

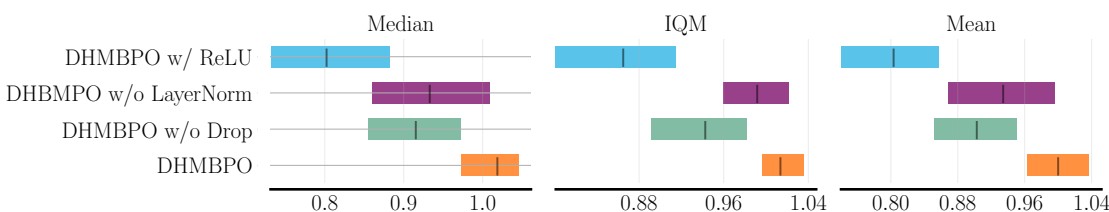

(a) Sample efficiency

(b) Critic loss

Figure 21: Ablation study of model architecture on sample efficiency and critic loss. The graphs compare DHMBPO with three variants about its model: (DHMBPO w/o LayerNorm) a model without LayerNorm, (DHMBPO w/o Dropout) a model without Dropout, and (DHMBPO w/ ReLU) a model in which all activation functions were changed from SiLU to ReLU.

Figure 22: Aggregation metrics about the final test return in the ablation study of model architecture.

### D.7 Distribution of predicted rewards and target values for critic

We applied the MBPO algorithm HalfCheetah using the MBRL-lib and created histograms of batch size rewards and critic losses calculated from the rollout buffer sampling every 10,000 interaction steps. We repeated this process up to 50,000 steps and summarized the results in Figure 23. While there are only a few outliers that can be considered as extreme values for rewards, the target values exhibit more variability, occasionally resulting in many outliers. Considering that gamma is 0.99, it can be stated that the samples at the 30,000th and 50,000th steps are extreme valued samples.

Although the causal relationship between the outliers in rewards and the anomalies in target values is not clear, it is speculated that the target values start to exhibit variability earlier than the rewards. From this observation, bounding the target critics' outputs described in Section A.5 is suggested as an effective remedy. Additionally, since the reward value can take extreme value, we use 1-st and 99-th percentiles of rewards for estimation of the lower bound and the upper bound, instead of the minimum and the maximum.

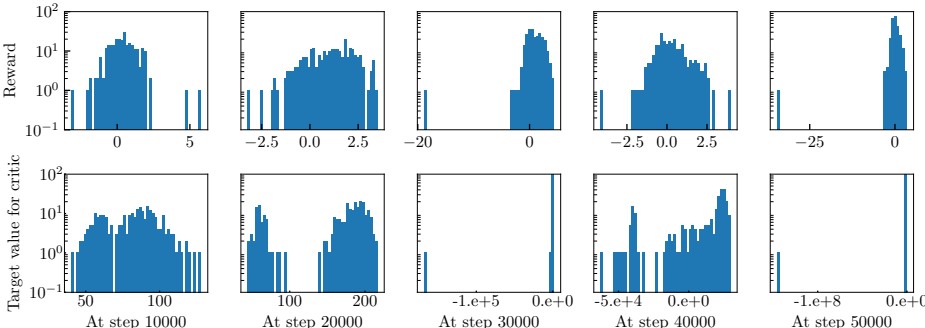

Figure 23: The transition of histograms for rewards (upper row) and target values (lower row) at every 10,000 environment steps during the execution of the MBPO algorithm on the "HalfCheetah" task. In each panel, the x-axis represents the value, and the y-axis represents the count. Noticeable outliers occur in the target values at the 30,000th and 50,000th steps.

