# OpenReview forum: "Double Horizon Model-Based Policy Optimization"
_TMLR — Accepted by TMLR_

### Review · Reviewer_UNGT · 2025-02-27

**Summary Of Contributions:**

Choosing a proper rollout length in model-based RL can be a challenging problem. To address it, this paper proposes a double-horizon scheme that uses long rollouts to learn the model and short rollouts to update the actor and critic network.

**Audience:**

Yes

**Broader Impact Concerns:**

None.

**Claims And Evidence:**

Yes

**Requested Changes:**

There are a few minor changes I would suggest:

+ In section 2.2, line 8, an equal sign may be mistakenly inserted after "Following existing works";

+ The objective function of model learning is now located in the appendix, which should be moved to the main text if space allows;

+ Ablation studies on the network architecture (e.g., the use of layer norm, dropout rate and activation function) would further strengthen the paper.

**Strengths And Weaknesses:**

Strengths:

+ The problem studied in this paper is well-motivated and important. I believe that choosing the rollout length is one of the most fundamental problems in RL.

+ This paper conducts extensive and comprehensive experiments to test various aspects of their algorithm.

+ The proposed algorithm performs well compared to existing methods in terms of overall return and sample efficiency.

+ The presentation is clear and easy to follow.


Weakness:

+ I am unsure how the actor objective given in (14) works. According to equation (7), differentiating both sides w.r.t. the policy parameter $\theta$ yields $ \nabla_\theta \hat{V} (s) = -\sum_{t=0}^T \gamma^T \alpha \nabla_\theta \log \pi_\theta(s_t|a_t) $ since other terms are independent of $\theta$, which is quite different from the DDPG objective described in (12) that differentiates through the Q network. I checked with the attached code and it differentiates every action $a_t$, i.e., treating $a_t$ as a function of $\theta$ instead of just a sample point as in policy gradient methods. This could be a critical gap between the methodology presented in the paper, and the implementation leading to the empirical results. I hope that the authors can either correct my understanding or explain this gap.

---

> ### Author Response · Authors · 2025-03-23
> **Response to Reviewer UNGT**
>
> >I am unsure how the actor objective given in (14) works. According to equation (7), differentiating both sides w.r.t. the policy parameter θ yields ∇θV^(s)=−∑t=0TγTα∇θlog⁡πθ(st|at) since other terms are independent of θ, which is quite different from the DDPG objective described in (12) that differentiates through the Q network. I checked with the attached code and it differentiates every action at, i.e., treating at as a function of θ instead of just a sample point as in policy gradient methods. This could be a critical gap between the methodology presented in the paper, and the implementation leading to the empirical results. I hope that the authors can either correct my understanding or explain this gap.
>
> Thank you for reviewing the code.
>
> The reviewers' main interest seems to be not in Equation (14) itself, but rather in its gradient with respect to $\theta$. The computation of this gradient follows the same approach as for the gradient of Equation (10). Specifically, we compute the value gradient field using a combination of the RP Trick for both state and action and backpropagation through time.
>
> Since both the state $s$ and reward $r$ are outputs of neural networks, computing the gradient of Equation (7) with respect to $\theta$ via backpropagation results in summing gradients along a single trajectory. The explicit formulation can be found in Equation (4.1) of [Zhang et al.].
>
> To clarify the relationship with TR-based methods, we have moved the definition of the value gradient field to Section 2.4 and revised Section 3 to explicitly state that the gradient computation follows the same procedure as TR-based methods.
>
> >In section 2.2, line 8, an equal sign may be mistakenly inserted after "Following existing works";
>
> Thank you for pointing this out. We have removed the equal sign.
>
> >The objective function of model learning is now located in the appendix, which should be moved to the main text if space allows;
>
> We added a concise introduction of the model we used, with guidance to appendix detailing about the model
>
> >Ablation studies on the network architecture (e.g., the use of layer norm, dropout rate and activation function) would further strengthen the paper.
>
> We added the results of an ablation study on the network architecture of the dynamics and reward models to Figures 21 and 22 in Appendix D.5. Specifically, we evaluated three variants: (1) a model without LayerNorm, (2) a model without Dropout, and (3) a model in which all activation functions were changed from SiLU to ReLU.
>
> From the results in Figure 21, we observed that removing LayerNorm caused the critic loss to diverge and led to a decline in policy performance. For the other two variants, the critic loss did not diverge; however, as shown in the aggregation metrics in Figure 22, their performance significantly deteriorated.
>
> While MBPO also uses SiLU, and thus this is not a novel contribution of our work, the reviewers' comments helped us highlight key factors in model training and reinforcement learning.

---

> > ### Comment · Reviewer_UNGT · 2025-03-27
> >
> > Thanks for the detailed response, my concerns are now addressed.

---

> > > ### Author Response · Authors · 2025-03-28
> > > **Response to Reviewer UNGT**
> > >
> > > We appreciate the response, and are glad that your concerns were addressed.

---

### Review · Reviewer_CSUu · 2025-03-10

**Summary Of Contributions:**

This work contributes a novel combination of strategies for reinforcement learning which are then shown to be highly efficient in learning RL tasks, especially when compared in run-time (wall-clock). This combination is motivated by expressing the fundamental trade-offs in selecting one's distribution rollout length or training rollout length and how these two processes can be combined with a 'double horizon' such that each has an optimised rollout length for better training of the actor and critic models. Results show a significant benefit of the proposed Double Horizon Model-Based Policy Optimization (DHMBPO) method over alterative, even state-of-the-art models, particularly in runtime though also in ease of application.

**Audience:**

Yes

**Claims And Evidence:**

Yes

**Requested Changes:**

None. I do believe that this paper is well written, concise, and quite honest about it's contribution. I am not an expert in this particular sub-field, but should there be no fundamental issues raised by the other reviewers, I would be inclined towards a straightforward accept.

**Strengths And Weaknesses:**

**Strengths**
- Well written paper and extremely easy to follow
- Strong results and good comparison against existing models as well as ablations

**Weaknesses**
- Largely this is an incremental contribution, with little deep mathematical or technical insight. Nonetheless the results are very impressive.

---

> ### Author Response · Authors · 2025-03-23
> **Response to Reviewer CSUu**
>
> Thank you for your review and positive comments.

---

### Review · Reviewer_xjto · 2025-03-13

**Summary Of Contributions:**

This paper studies model-based policy gradient and proposed Double Horizon Model-Based Policy Optimization (DHMBPO), which performs policy gradient on long distribution rollouts and short training rollouts.

**Audience:**

Yes

**Claims And Evidence:**

Yes

**Requested Changes:**

1. Can the authors discuss the distinctions or add experimental comparisons between [1, 2]?
2. More clearly motivate the naming of the method.
3. (Optional) Add model-free baselines to experiments.

**Strengths And Weaknesses:**

Strengths:
1. The paper is clearly written and includes enough details, such as their motivation and how their method relates to previous methods.
2. The experiments and ablation studies also support the claims well.

Weaknesses:
1. The novelty of the proposed method is unclear. It is not obvious how the proposed method is different from [1, 2]. Specifically, could the authors elaborate on the difference between DHMBPO and the 'Model Derivatives on Predictions' variant in [1], which was originally proposed in [2]?
2. The name Double Horizon Method is somewhat confusing and lacks an explanation.
3. The experiments can be further enhanced by comparing them with model-free baselines.

[1] Zhang et al. "Model-Based Reparameterization Policy Gradient Methods: Theory and Practical Algorithms."\
[2] Clavera et al. "Model-Augmented Actor-Critic: Backpropagating through Paths."

---

> ### Author Response · Authors · 2025-03-23
> **Response to Reviewer xjto**
>
> Thank you for the review.
>
> >1. Can the authors discuss the distinctions or add experimental comparisons between [1, 2]?
>
> The difference between DHMBPO and [1] or [2] lies in whether the initial state distribution for computing Derivatives on Predictions (DP) originates from the true environment or from model predictions. This can also be seen as the presence or absence of distribution rollout.
> (For clarity about the term DP, this is how [2] refers to backpropagation through time with model predictions and a terminal critic for computing the value gradient.)
>
> DHMBPO is an algorithm that encompasses [1] and [2], and estimates the policy gradient using DP. However, in DHMBPO, the initial states for DP computation are randomly sampled from the model buffer, as adopted in the MBPO implementation by Janner et al. (2019). As described in Section 2.3, this buffer consists of states generated by model predictions.
>
> On the other hand, in the mathematical formulation of [1], DP computation employs a mixture of the initial state distribution and the state-action visitation measure $\sigma_\pi(s, a) (1 - \gamma) \sum_{i=0}^\infty \gamma^i \Pr(s_i=s, a_i = a)$, where $\Pr(s_i=s, a_i = a)$ is a probability the $(s, a)$ is observed at step $i$ under control of policy $\pi$. The actual algorithm randomly draws samples from states within finite-step on-policy trajectories. Either way, the state distribution is not from model predictions.
>
> These differences are illustrated in Figure 1. Methods [1] and [2] correspond to the "TR-based" approach indicated in green in the figure, while the proposed DHMBPO method is shown in pink, following a different computational process.
>
> Additionally, for experimental comparisons on GYM tasks with [1], we have included RP-PGM in Figure 2. We used the author’s open-sourced implementation. The accompanying explanation has been added to Section 4.2 and Appendix B.
>
>
>
> >2. More clearly motivate the naming of the method.
>
> This study proposes executing rollouts with different horizons for TR, i.e., rollouts for computing DP, and for DR, i.e., rollouts for storing data in the model buffer. Since two rollouts with different horizons are performed, we named our method "Double Horizon."
>
> Reflecting this distinction, we introduced the name **Double Horizon Method** in the latter part of the introduction. This revision makes the core idea of our approach more comprehensible.
>
> >3. (Optional) Add model-free baselines to experiments.
>
> We included SAC, a model-free algorithm that optimizes the entropy-regularized cumulative reward, as a baseline for comparison alongside DHMBPO, MBPO, and SVG-SAC(H), all of which also maximize the entropy-regularized cumulative reward.
>
> We updated Figures 2 and 3, as well as Figures 8 and 9 in Appendix C.2, to incorporate SAC in the comparative experiments on GYM and DMC tasks. As described in the revised Appendix B, the SAC results are based on our implementation. Additionally, as shown in the revised Table 2 in Appendix A.4, the hyperparameters for SAC were set to be consistent with those of DHMBPO.
> We think this comparison provides a meaningful implication.

---

> > ### Comment · Reviewer_xjto · 2025-03-27
> >
> > Thank the authors for the detailed responses, which address my previous concerns. I believe the current version of the manuscript that incorporates the above clarification can better help readers understand the novelty and its distinctions from previous model-based methods. For the model-free baselines, I would recommend that the authors add a horizontal line that indicates the asymptotic performance of SAC, as they typically need more steps to converge.

---

> > > ### Author Response · Authors · 2025-03-28
> > > **Response to Reviewer xjto**
> > >
> > > We appreciate your thoughtful feedback and are glad to hear that our revisions have addressed your previous concerns. Regarding your suggestion to include a horizontal line indicating the asymptotic performance of SAC, we agree that this would improve the clarity of the results. We will incorporate this addition into the camera-ready version of the paper. Based on previously published results, the asymptotic performance of DHMBPO generally exceeds that of SAC.

---

> > > > ### Author Response · Authors · 2025-03-31
> > > > **Additional Response to Reviewer xjto**
> > > >
> > > > The experiments to measure SAC's asymptotic performance have now been completed. Thus, we incorporated these results into the revised manuscript. Specifically, we conducted additional experiments using SAC (5 seeds per task), running it for ten times the maximum environment step budget used for evaluating the model-based methods, to estimate its asymptotic performance. Figure 2.a and Figure 8 (Appendix C.2) have been updated to include horizontal reference lines indicating SAC's average test return at these extended environment steps. Additional explanations in the modified manuscript are indicated by blue-colored texts.

---

### Author Response · Authors · 2025-03-23
**Response to all reviewers**

Thank you for your valuable time. To address the review comments we have performed additional experiments to provide more direct evidence about the benefits of our implementation details (more on this later). In the revised manuscript, we highlight all modified texts in red to clearly indicate changes made during the review process. Note that coloring via the hyperref package has been disabled to maintain readability. Our contribution is now clearer.

Firstly, we want to repeat the strongest point of our work: the key advantage of our proposed method lies in its ability to achieve sample efficiency and shorter execution time comparable to or better than existing state-of-the-art (SOTA) methods on both GYM and DMC benchmarks without requiring any task-specific hyperparameter tuning.

Specifically, on the GYM benchmark, our method achieves comparable sample efficiency to MACURA in four out of five tasks and significantly outperforms it in the Walker2d task, attaining approximately twice the cumulative reward at 500K environment steps (our method also has several times faster computation). On the DMC benchmark, our method achieves approximately 30% shorter execution times compared to representative SOTA methods such as Dreamer v3 and TD-MPC2 (see Appendix C.2, Table 3 for detailed results).

Conventionally, algorithms achieving SOTA performance on the GYM benchmark have required task-specific hyperparameter tuning or complex scheduling, which hinder their practical applicability to new tasks. On the other hand, leading algorithms demonstrating high performance on the DMC benchmark (e.g., Dreamer v3 and TD-MPC2) successfully utilize common hyperparameters but are limited to latent-variable models.

In contrast, our study demonstrates, for the first time, that even feedforward neural network-based models—without latent variables—can achieve robust performance across tasks without task-specific hyperparameter tuning. By evaluating our method consistently on both GYM and DMC benchmarks, which are widely recognized as critical evaluation suites in the field of deep reinforcement learning, we clearly establish the practicality and versatility of our proposed approach. We consider this an important contribution of our work.

Moreover, we will release our code upon publication to enable maximum benefit to the community.

---

### Comment · Editors_In_Chief · 2025-08-27

On August 27, the EiCs updated the camera ready PDF and abstract, upon request of the authors. This change includes the Github URL in the abstract, and the PDF now includes acknowledgments.

---

### Decision · Action_Editor_M9P6 · 2025-04-18

**Recommendation:** Accept as is

**Comment:**

The paper suggests a new model-based reinforcement learning algorithm that combines ideas from two somewhat different approaches in using a model.
In one approach, one rolls out the model for multiple step in order to generate the distribution over states according to the current policy. One can then use the fictitious/imagined samples to update the critic or policy.
Another way is to use the model to expand the value function for multiple steps, and then compute the policy gradient by differentiating through the expanded value function and the model.

The main contribution of this work is proposing Double Horizon Model-Based Policy Optimization (DHMBP), which combines these two approaches of using the rolled out model. The paper empirically studies the proposed method, and compares with other model-based approaches, as well as a model-free one.


The reviewers are generally positive about the paper. Among the positive aspects, they mentioned:

- Clear and easy to follow presentation
- Strong experimental results

They were some initial concerns such as unclear novelty of the paper compared to some prior work and the lack of model-free baseline. These were clarified during the rebuttal phase and a model-free baseline was added in the revised version.

At the end, the reviewers are satisfied with this paper, and the final recommendation of all of them are an Accept (3x). Therefore, **I recommend the acceptance of this paper**.


I'd like to add some final comments based on my own reading of the paper. I think the paper has a reasonable idea. Since the paper does not have much of theoretical studies, the burden of its validity is mostly on the empirical results. The empirical results show good performance of the method. One aspect, however, is missing in my opinion, and that is a careful study of the role of each of the rollout parameters. The paper currently sets the DR horizon to 20 steps and the TR horizon to 5 steps throughout most experiments, except those where these values are set to the extremes of (20,0) and (0,5). The paper would be stronger if there are some studies of the effect of each of these parameters through more gradual changes. I realize this experiment is computationally expensive, so I think doing it for a limited set of experiments is acceptable. Such a result would give a better understanding of the role of each of these central hyper-parameters of the method.
This is an optional suggestion and is not a requirement for the acceptance of the camera ready version.


Typo:
- In Section 2.5, right after (12), "... by derived by extending ..." requires a revision.

**Audience:**

Yes. This paper would be of interest to reinforcement learning researchers, especially those who are interested in model-based methods.

**Claims And Evidence:**

Yes, the paper provides sufficient and clear evidence. Please read my comments for more detail.

---

> ### Author Response · Authors · 2025-04-25
> **Response to Action Editor**
>
> We sincerely appreciate your prompt response and the careful consideration you, as the Action Editor, have given to our work, which led to the recommendation for acceptance. We are grateful for the time and effort you devoted to reviewing our manuscript.
>
> We would also like to express our appreciation to the three anonymous reviewers for their constructive comments and insightful suggestions throughout the review process. Their feedback has been invaluable in improving the clarity and quality of our work.
>
> In response to your comment, we would like to offer a brief clarification. As you rightly pointed out, our paper emphasizes empirical validation rather than theoretical analysis. In particular, as shown in Figure 5 of Section 4.4, we conducted a hyperparameter search over the rollout lengths of both DR and TR. We believe this serves as an empirical examination of the respective roles of these rollout parameters. While the ablation study in Section 4.3 also presents results for the cases where DR = 0 and TR = 0, the initial paragraph of Section 4 did not explicitly mention the sensitivity analyses in Section 4.4.
>
> Given that both the experiments in Sections 4.3 and 4.4 aim to investigate the functional contributions of DR and TR, we have revised the first paragraph of Section 4 to clearly state this connection.
>
> We have submitted the updated manuscript with these and other minor revisions as our camera-ready version.

---

> > ### Comment · Action_Editor_M9P6 · 2025-04-26
> >
> > Thank you for your clarification! Yes, I missed the results in Section 4.4. Those are already addressing my suggestion.